# 🦉 PeRL: Permutation-Enhanced Reinforcement Learning for Interleaved Vision-Language Reasoning

**Yizhen Zhang**$^{\phi\pi*}$    **Yang Ding**$^{\phi*}$    **Shuoshuo Zhang**$^{\phi\pi*}$
**Xinchen Zhang**$^{\phi}$    **Haoling Li**$^{\phi\pi}$    **Zhong-Zhi Li**$^{\rho\pi}$    **Peijie Wang**$^{\rho}$
**Jie Wu**$^{\phi\pi}$    **Lei Ji**$^{\pi}$    **Yeyun Gong**$^{\pi\dagger}$    **Yelong Shen**$^{\pi}$    **Yujiu Yang**$^{\phi\dagger}$
$^{\phi}$Tsinghua University    $^{\pi}$Microsoft    $^{\rho}$CASIA
https://github.com/alchemistyzz/PeRL

## Abstract

Inspired by the impressive reasoning capabilities demonstrated by reinforcement learning approaches like DeepSeek-R1, recent emerging research has begun exploring the use of reinforcement learning (RL) to enhance vision-language models (VLMs) for multimodal reasoning tasks. However, most existing multimodal reinforcement learning approaches remain limited to spatial reasoning within single-image contexts, yet still struggle to generalize to more complex and real-world scenarios involving multi-image positional reasoning, where understanding the relationships across images is crucial. To address this challenge, we propose a general reinforcement learning approach PeRL tailored for interleaved multimodal tasks, and a multi-stage strategy designed to enhance the exploration-exploitation trade-off, thereby improving learning efficiency and task performance. Specifically, we introduce permutation of image sequences to simulate varied positional relationships to explore more spatial and positional diversity. Furthermore, we design a rollout filtering mechanism for resampling to focus on trajectories that contribute most to learning optimal behaviors to exploit learned policies effectively. We evaluate our model on 5 widely-used multi-image benchmarks and 3 single-image benchmarks. Our experiments confirm that PeRL trained model consistently surpasses R1-related and interleaved VLM baselines by a large margin, achieving state-of-the-art performance on multi-image benchmarks, while preserving comparable performance on single-image tasks.

## 1 Introduction

Real-world applications often require the ability to understand and reason over multiple images—a capability that humans naturally possess. In response, several multi-image benchmarks (e.g., [14, 19, 35, 46]) have been introduced to support the study of such comprehensive and practically relevant tasks within the research community. Specifically, multi-image multimodal benchmarks span a wide range of tasks involving diverse multimodal relationships, including: 1) Low-level visual features, such as comparing illumination, color saturation, or texture; 2) High-level semantic relationships, such as emotional associations or conceptual similarities between objects; 3) Temporal reasoning, involving the understanding of event sequences or changes over time; 4) 2D/3D spatial relationships, including rotation, symmetry, and variations in camera viewpoints. Furthermore, the interleaved format of images and text, as introduced in works like [3], provides a unified and flexible template that

---

*Equal contribution. Work done during the internship of Yizhen, Shuoshuo, Haoling, Zhong-Zhi and Jie at Microsoft Research.✉ :zhangyizhen24@mails.tsinghua.edu.cn

†Corresponding authors.

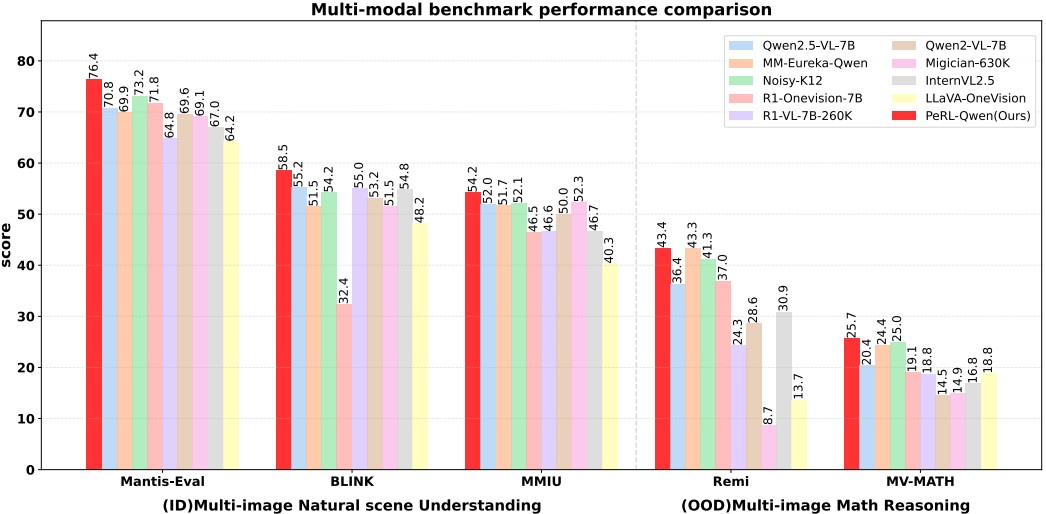

Figure 1: Compare PeRL against baseline approaches, including instruction-tuned or R1-like reinforcement learning models, across both in-domain and out-of-domain multi-image benchmarks.

supports both single-image and multi-image scenarios. This format allows for arbitrary arrangements of textual and visual inputs, enabling dynamic and context-aware interactions that more closely resemble real-world reasoning tasks.

Despite the impressive progress of vision-language models (VLMs) on single-image tasks, their ability [16, 53, 54, 5] to understand and reason over multiple images remains significantly underexplored. Motivated by this gap, recent works such as [19, 22, 24] have constructed multi-image instruction-tuning datasets to further post-train general-purpose VLMs for multi-image tasks. In parallel, models like Flamingo [3], LLaVA-Interleave [22], Migician [24] and Mantis-Idefics [19] have been trained on interleaved image-text data to enhance their multi-image understanding capabilities. However, despite these efforts, interleaved VLMs still struggle with reasoning across images, as highlighted in [45]. Their limitations become especially apparent in tasks that require positional alignment, temporal sequencing, or holistic cross-image understanding. Meanwhile, reinforcement learning (RL) has recently emerged as a promising direction for enhancing multimodal reasoning in VLMs [34], showing encouraging results on single-image tasks. Nonetheless, these RL-augmented models also fall short when applied to more complex multi-image scenarios, particularly those demanding deep reasoning across spatial and semantic contexts.

Concretely, multi-image scenarios introduce challenges in both spatial reasoning—understanding object relationships within a single image—and positional reasoning—understanding relationships across multiple images. As illustrated in Fig. 2, simply altering the order of input images can lead to incorrect predictions from both general-purpose and instruction-tuned vision-language models. For instance, the general model Qwen-2.5-VL-7B [7] fails to distinguish the visual attributes of each image independently, incorrectly referencing "bright yellow and blue hues" for both images. Meanwhile, the instruction-tuned model Migician [24] correctly identifies the "red and green pattern" but misattributes it to the wrong image, indicating a failure in positional alignment. These examples highlight the limitations of current VLMs in maintaining consistent and grounded reasoning across interleaved multi-image inputs.

To address these challenges, we explore interleaved multimodal reasoning through reinforcement learning and design a simple yet effective strategy to mitigate positional bias. Specifically, we develop a multi-stage data processing pipeline that generates diverse training samples by permuting the order of input images. Since such permutations may alter the semantic alignment between text and images, we correspondingly rephrase the textual input to maintain semantic consistency. However, we observe a significant imbalance in the difficulty levels of the resulting training samples leading to inefficient reinforcement learning. To mitigate this, we introduce a rollout filtering mechanism which helps stabilize training and improve learning efficiency. This approach allows our proposed model,

Qwen-PeRL, to focus on informative trajectories and better generalize across complex multi-image reasoning tasks.

We summarize our main contributions as follows:

- We propose PeRL, a simple yet effective strategy that permutes image orders and rephrases text to address multi-image coreference, while a rollout filtering mechanism improves training efficiency by focusing on informative samples.

- As shown in Fig. 1, extensive experiments on both in-domain and out-of-domain benchmarks demonstrate that our method achieves state-of-the-art performance on multi-image tasks and competitive results on single-image tasks.

## 2 Related Works

### 2.1 Multimodal Reasoning

Early efforts in multimodal reasoning [36, 56, 25, 41, 4, 26] mainly focus on supervised fine-tuning (SFT) especially the dataset construction with chain-of-thought (CoT) [28, 27, 61] reasoning steps. Recently, Deepseek-R1-Zero [15] demonstrates remarkable reasoning capabilities, spearheading large-scale reinforcement learning (RL) research in LLM. While VLMs also require substantial reasoning capabilities for vision-and-language tasks, recent investigations employ reinforcement learning to multimodal reasoning and present early progress including URSA [33], Visual-RFT [31], LMM-R1 [37], VLM-R1 [39], Reason-RFT [43], MM-EUREKA [34], R1-OneVision [55], Vision-R1 [17], R1-V [8], R1-VL [58], VL-Rethinker [47], VisualThinker-R1-Zero [62], OpenVLThinker [12] and Perception-R1 [57]. Most of these works adopt GRPO [38] for RL scaling through *data* construction [43, 17, 34, 62] or mixture [37], multi-stage *training* strategy [37, 12, 17, 34] or perception-wise [31, 57, 39] or step-wise [58] *reward* design . Distinct from these approaches, VL-Rethinker [47] strategically select high-value samples to avoid vanishing advantages, while NoisyRollout [30] introduces moderately distorted images as diverse samples for broad exploration to address the challenge of imperfect visual perception.

### 2.2 Interleaved VLM

With the significant advancement in single-image tasks, vision-language models (VLMs) [1, 44, 2, 10, 52, 7, 52, 23, 11, 18] have emerged, *pretraining* on interleaved image-text datasets [48] to address more comprehensive multi-image tasks in real-world scenarios. Building on this, [23, 19, 24] further construct interleaved multi-image-text data to *post-train* VLMs with instruction tuning. These approaches either exploit learnable cross-attention modules with a Perceiver Resampler [3, 6, 21], or utilize a linear projection layer [22, 42] to integrate pretrained language models with visual encoders. Nonetheless, multi-image tasks still face further challenges beyond the perception problem in single-image tasks, including complex spatial and positional/temporal reasoning. To address the bias, [45] proposes SoFt Attention (SoFA), a simple, training-free method to mitigate the position bias of VLMs when processing multiple images. Differently, we propose to scale the interleaved VLM with reinforcement learning to incentive general multimodal reasoning capability and further design a comprehensive rollout strategy for coreference resolution.

## 3 Problem Formuation and Analysis

The multimodal understanding task is defined as predicting an output answer given a text query interleaved with images. Formally, an input instance is represented as $x = (I, Q)$, where $Q$ denotes the textual query and $I = \{i_j\}_{j=1}^n$ is the associated set of images. The corresponding output is denoted by $y$, yielding a dataset $D = \{(x_i, y_i)\}_{i=1}^N$. In practice, we interleave text tokens with image representations using a special placeholder token <image> to indicate the presence and position of an image within the input sequence.

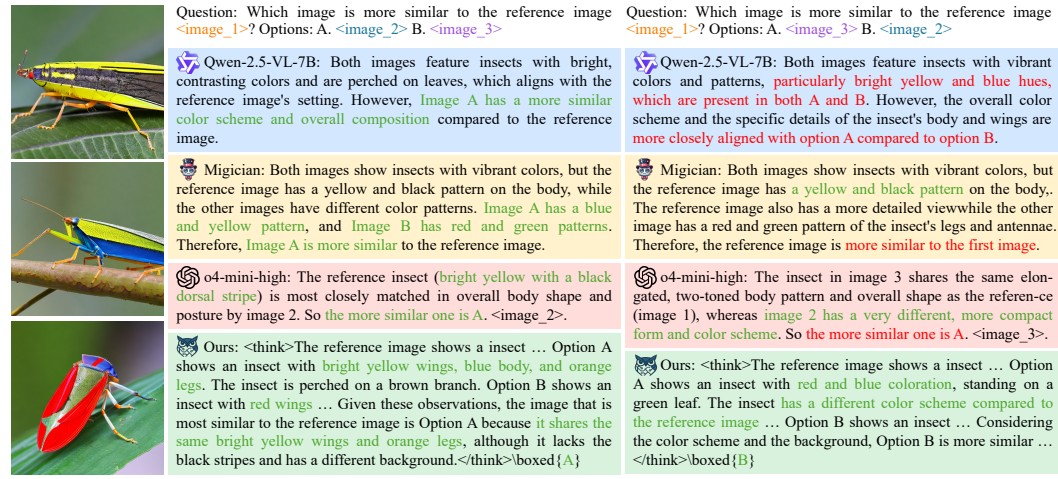

Figure 2: A showcase highlighting the challenges faced by existing VLMs: 1) Fail to recognize or reason across multiple images; 2) VLMs exhibit positional bias, leading to inconsistent answers when the order of images is permuted even though the semantic of the input keeps the same.

## 3.1 Positional Bias

One of the critical challenges in multi-image multimodal reasoning is positional coreference, which requires accurately identifying the specific image that corresponds to a given textual reference. However, as highlighted in [45], existing vision-language models (VLMs) still suffer from significant performance degradation when the order of input images is altered. As illustrated in Fig. 2, we summarize the major limitations of current VLMs in handling multi-image tasks. Motivated by these observations, we propose a simple yet effective strategy that diversifies image inputs through permutation, while simultaneously modifying the associated text to preserve semantic invariance. This approach encourages the model to develop a more robust understanding of positional relationships across images, thereby improving generalization in complex multimodal reasoning scenarios.

## 3.2 Difficulty Imbalance

Mantis-Instruct [19] is the first multi-image instruction-tuning data, comprising 721K data. Therefore, we start with the Mantis-Instruct dataset as our training data. Before training, we firstly exploit the Qwen-2.5-VL [7] model to generate several outputs and calculate the average accuracy among these generations as the difficulty score of each sample for statistical analysis. Fig. 3b represents the difficulty distribution. We find that the distribution is extremely imbalanced, leading to inefficient and unstable reinforcement learning. Inspired by this finding, we further resample the data for a balanced difficulty distribution. In the following section, we discuss the pipeline to process the data according to the two observations.

## 3.3 Data Prepocessing

We develop a multi-stage data processing pipeline that systematically filters and reformulates QA pairs from the original training corpus Mantis-Instruct [19] to construct a higher-quality training subset, as illustrated in Fig. 3a.

Initially, we apply a rule-based filter to the original dataset, screening entries based on question types, image quantity, and other relevant attributes. Then we employ GPT-4o [1] to perform format rephrasing on open-ended questions that are challenging to verify through rule-based methods. In the next phase, we implement a rollout filtering mechanism using Qwen2.5-VL-7B [7]. For each question, we perform rollout 10 times and calculate the average accuracy of the responses. Questions are then filtered based on their accuracy score, resulting in a refined distribution as shown in Fig. 3b.

To evaluate the semantic sensitivity of questions to image order, we introduce a semantic variation analysis step using GPT-4o. Specifically, as illustrated in Fig. 3a, given an input–output pair $(x, y)$,

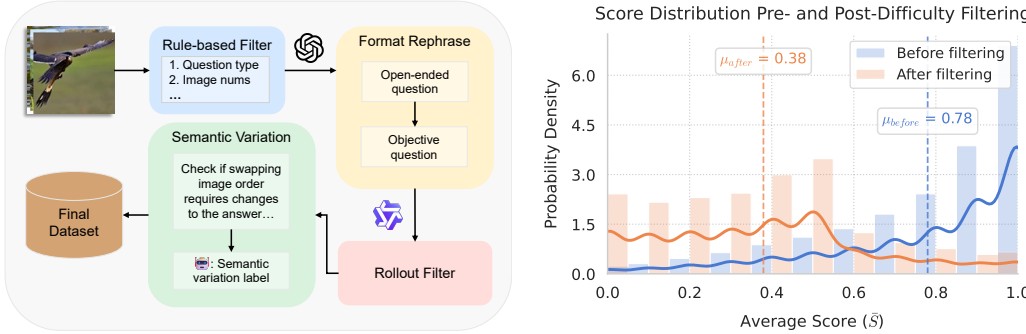

(a) Multi-Stage Preprocessing Pipeline  (b) Score Distribution Pre- and Post-Difficulty Filtering

Figure 3: Overview of Data Preprocessing and Difficulty Distribution. (a) The multi-stage pipeline applies (i) rule-based filtering, (ii) format rephrasing, (iii) rollout-based filtering and (iv) semantic variation checking to curate QA pairs. (b) KDEs of the average score $\bar{S}$ before (blue) and after (orange) rollout-based difficulty filtering, with the mean falling from 0.78 to 0.38.

we apply a random permutation $\sigma$ to reorder the image set, resulting in a new input $\hat{x} = (\hat{I}, Q)$ with $\hat{I} = \{i_{\sigma(j)}\}$. GPT-4o is then prompted to assess whether the permuted input $\hat{x}$ preserves the semantics of the original input $x$, i.e., whether the intended answer should remain unchanged. The corresponding output $\hat{y}$ is defined as:

$$\hat{y} = S(x, \hat{x}) \cdot y + (1 - S(x, \hat{x})) \cdot \Lambda(y, \sigma) \tag{1}$$

Here, $S(x, \hat{x}) \in \{0, 1\}$ is a semantic equivalence indicator obtained by prompting GPT-4o, where $S = 1$ denotes the input semantics are preserved under permutation. The function $\Lambda(y, \sigma)$ denotes the transformation applied to the answer when semantic equivalence does not hold. It adapts $y$ to the permuted context according to the structure of the answer space $Y$, such as reindexing visual choices.

This framework facilitates controlled perturbation of the visual input via image sequence permutations, aiming to simulate diverse spatial and temporal configurations and thereby encourage richer generalization and reasoning capabilities.

## 4 Methods

The goal of multimodal model is to train a policy model $\pi_\theta(y|x)$ parameterized by $\theta$ to optimize the correctness of model response $o$. In this paper, we exploit reinforcement learning (RL) framework specifically GRPO to learn to improve the multimodal reasoning as presented in Fig.4. Firstly, the policy model $\pi_\theta(y|x)$ takes the input $x$ to generate a group of responses $\{o_i\}$. Next, the reward model calculates the score $r(x, o_i, y)$ for each response $o_i$, which adopts rule-based functions to evaluate the accuracy and format correctness. Furthermore, the group computation module estimates the advantage $A_i$ for each response $o_i$ to update the policy model.

Specifically, we propose a plug-and-play module to improve the efficient and effective training through diversified advantage estimation and dynamic resampling. 1) Before exploration, we resample rollouts with variable reward distribution to exploit learned policies effectively. This ensures the learning focuses on trajectories that contribute most to learning optimal behaviors across both spatial and positional dimensions.2) To mitigate positional bias in interleaved multi-image reasoning, we introduce diversity in the rollout phase: for Single-image,we keep the original format. for multi-image positional diversity, use permutations of image sequences to simulate varied positional relationships.

### 4.1 GroupRelative Policy Optimization

GRPO [38] estimates the advantage score based on the reward normalization of a group of responses given a specific query. Particularly, the advantage score $A_i$ and the corresponding loss are defined as:

$$A_i = \frac{r(x, o_i, y) - \frac{1}{|o|} \sum_{j=1}^{|o|} r(x, o_j, y)}{\sqrt{\frac{1}{|o|} \sum_{j=1}^{|o|} \left( r(x, o_j, y) - \frac{1}{|o|} \sum_{j=1}^{|o|} r(x, o_j, y) \right)^2}} \tag{2}$$

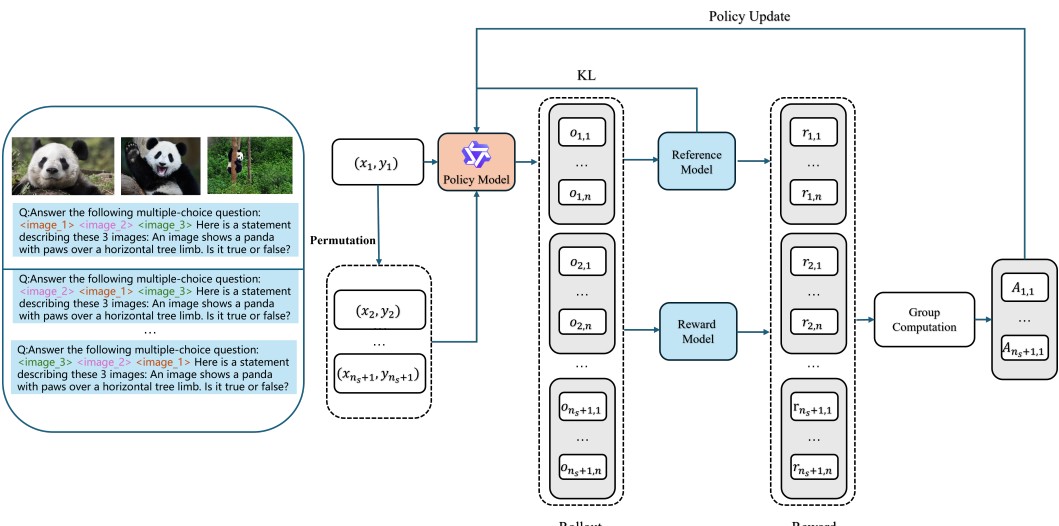

Figure 4: An illustration of PeRL framework. $(x_1, y_1)$ denote original input sample, while $\{(x_i, y_i)\}_{i=2}^{n_s+1}$ represent samples generated through permutation operation, $n_s$ is the number of permutations per sample and $n$ is the number of rollouts per order. $o_{i,k}, r_{i,k}$ and $A_{i,k}$ represent the trajectory, reward and advantage of $k_{th}$ rollout of sample $(x_i, y_i)$ respectively.

$$\mathcal{L}_{GRPO} = \frac{1}{|o|} \sum_{i=1}^{|o|} \left( \min \left( \frac{\pi_\theta(o_i|x)}{\pi_{old}(o_i|x)} A_i, \text{clip}(\frac{\pi_\theta(o_i|x)}{\pi_{old}(o_i|x)}, 1 - \epsilon, 1 + \epsilon) A_i \right) - \beta \mathcal{D}_{KL}(\pi_\theta||\pi_{ref}) \right) \tag{3}$$

where $|o|$ is the group size, $o_i$ is the $i_{th}$ response given the input $x$. $\pi_{old}$ is the old policy model to generate the response $o$, and $\pi_\theta$ is the current policy model to optimize. $r(x, o_i, y)$ is the rule-based reward function to calculate the accuracy and format correctness of the response $o_i$.

## 4.2 Permutation GRPO

Let $(x_1, y_1)$ denote the original input sample, where $x_1$ consists of a query and one or more images, and $y_1$ is the corresponding answer. We augment it in to a group of samples $\{(x_2, y_2), (x_3, y_3)...(x_{n_s+1}, y_{n_s+1})\}$. For multi-image input, we augment it by swapping images in $x$ randomly and modify the the answer $y$ correspondingly. To ensure training stability, we introduce a linearly decaying factor $\alpha_t$ (scheduled over training steps $t$) that controls the probability of applying image swapping per training batch. In the rollout progress, we input the diversified samples to the policy model to generate response $o_i$ and the reward model for further reward calculation. After that, we merge all $n_s + 1$ groups of responses together to calculate the baseline, final advantage and loss function as:

$$\bar{R} = \frac{1}{\sum_{i=1}^{n_s+1} |o_i|} \sum_{i=1}^{n_s+1} \sum_{k=1}^{|o_i|} r(x_i, o_{i,k}, y_i) \tag{4}$$

$$A_{i,k} = \frac{r(x_i, o_{i,k}, y_i) - \bar{R}}{\sqrt{\frac{1}{\sum_{i=1}^{n_s+1} |o_i|} \sum_{i=1}^{n_s+1} \sum_{k=1}^{|o_i|} (r(x_i, o_{i,k}, y_i) - \bar{R})^2}} \tag{5}$$

$$\mathcal{L}(\theta) = \frac{1}{\sum_{i=1}^{n_s+1} |o_i|} \left( \sum_{i=1}^{n_s+1} \sum_{k=1}^{|o_i|} \min \left( \frac{\pi_\theta(o_{i,k} \mid x_i)}{\pi_{\text{old}}(o_{i,k} \mid x_i)} A_{i,k}, \text{clip}(\frac{\pi_\theta(o_{i,k} \mid x_i)}{\pi_{\text{old}}(o_{i,k} \mid x_i)}, 1 - \epsilon, 1 + \epsilon) A_{i,k} \right)$$
$$- \beta \mathcal{D}_{\text{KL}}(\pi_\theta \parallel \pi_{\text{ref}}) \right)$$
$$\tag{6}$$

where $o_{i,k}$ denote the $k_{th}$ response for input $x_i$, and $A_{i,k}$ represent the advantage of this rollout. In our experiments, we set the number of responses per order $|o_i|$ to a same value, i.e. $|o_1| = |o_2| = \cdots = |o_{n_s+1}| = n$. Thus the actual group size is $\sum_{i=1}^{n_s+1} |o_i|$.

Note that unlike NoisyRollout [30], which performs policy update solely conditioned on the original images, we calculate policy loss of each rollout group $o_i$ based on augmented inputs $x_i$, as shown in Equation 6. This is because the answer is also updated after permutation, continuing to use the original prompt as a condition is improper and might lead to model collapse.

---

**Algorithm 1** PeRL: Permutation-Enhanced Reinforcement Learning

---

**Input:** Current policy $\pi_\theta$, old policy $\pi_{\theta_{old}}$, dataset $\mathcal{D}$, training steps $T_{\max}$, clip parameter $\epsilon$, initial decay factor $\alpha_0$, linear decay scheduler $\eta(\cdot)$, permutation operation $P(\cdot)$, permutation number $n_s$, rollout number per order $n$

**for** $t = 1$ to $T_{\max}$ **do**
    Sample batch $(\mathbf{x_1}, \mathbf{y_1}) \sim \mathcal{D}, \mathcal{B} = \{(\mathbf{x_1}, \mathbf{y_1})\}$
    Set decay factor $\alpha_t = \eta(\alpha_0, t, T_{\max})$
    **for** $i = 2$ to $n_s + 1$ **do**
        Generate randomly swapped images and modify answers correspondingly, $(\mathbf{x_i}, \mathbf{y_i}) = P((\mathbf{x_1}, \mathbf{y_1}), \alpha_t)$
        $\mathcal{B} \leftarrow \mathcal{B} \cup (\mathbf{x_i}, \mathbf{y_i})$
    **end for**
    **for** each sample $(\mathbf{x_i}, \mathbf{y_i})$ in $\mathcal{B}$ **do**
        Generate responses $\{\mathbf{o}_{i,k}\}_{k=1}^n$ from $\pi_\theta(\mathbf{o} \mid \mathbf{x_i})$
    **end for**
    Compute baseline $\bar{R}$ according to Eq.4
    Compute advantages $\hat{A}_i$ according to Eq.5
    Update policy according to Eq.6
    $\theta \leftarrow \theta - \nabla_\theta \mathcal{L}(\theta), \theta_{old} \leftarrow \theta$
**end for**

---

## 5 Experiment

**Training**   Our training data comprise two parts: 22K multi-image instruction examples curated from the 721K examples in Mantis-Instruct [19], and 36K single-image examples from the K12 dataset for RL. We initialize our policy with Qwen2.5-VL-7B-Instruct [7] and build on the veRL framework [40]. During RL fine-tuning, we apply one random permutation per sample ($n_s = 1$) and generate six responses per order ($n = 6$), yielding 12 rollouts per input. We set the KL coefficient $\beta = 0.01$, train for 2 epochs with a learning rate of $1 \times 10^{-6}$ and a batch size of 256. Further details are provided in the appendix.

**Benchmarks**   We conduct experiments on both multi-image benchmarks and single-image benchmarks. As the main experiment, we employ Mantis-Eval [19], BLINK [14], MMIU [35] as multi-image benchmarks. Furthermore, we evaluate the generalization on widely used single-image benchmarks including MathVista [32], MathVerse [60] and MathVision [49]. Besides, we also evaluate our model on out-of-domain multi-image benchmarks including Remi [20] and MV-Math [50]. We directly employ the VLMEvalKit [13] to evaluate the performance.

### 5.1 Main Results

**Baselines**   The compared models adopt different training strategies. LLaVA-v1.5-7B [29], LLaVA-NeXT-Interleave [23], LLaVA-OneVision [22], Qwen2-VL-7B [51], Qwen2.5-VL-7B [7], Deepseek-VL-7B [52], InternVL2 [11], and InternVL2.5 [9] are instruction-tuned base models. Migician-630K [24] is further fine-tuned on MGrounding-630K to enhance multi-image grounding. Models like LLaVA-OneVision-+VISC-150K [59] and Qwen2-VL-7B-VISC-150K [47] are fine-tuned on synthetic VISC-150K dataset, which explicitly encodes inter-image relations.

Table 1: Model performance across VQA and math reasoning benchmarks. Among Instruct- and RL-tuning models, **bold** numbers denote the best value in each column, and underlined numbers are the second-best. Task categories: multi-image natural scene understanding, single-image math reasoning, and multi-image math reasoning. The last column (AVG) is the mean of all eight metrics for rows with complete scores. Models marked with $^*$ are evaluated via vLLM inference.

| Model | Size | Mantis-Eval | BLINK | MMIU | MathVista | MathVerse | MathVision | Remi | MV-MATH | AVG |
|---|---|---|---|---|---|---|---|---|---|---|
| Human | – | – | 95.67 | – | 60.30 | 64.90 | 68.82 | 95.80 | 76.50 | – |
| Random | – | – | 38.10 | 27.40 | 17.90 | 12.40 | 7.17 | – | – | – |
| GPT-4o | – | 68.00 | 68.00 | 55.70 | 63.80 | 37.60 | 30.60 | 59.96 | 32.10 | – |
| *Multimodal General Models* | | | | | | | | | | |
| VILA-1.5 | 8B | 51.15 | 39.30 | 32.45 | 65.40 | – | – | – | – | – |
| Mantis-Idefics2 | 8B | 57.14 | 49.05 | 45.60 | 40.40 | 15.05 | 12.40 | – | 5.50 | – |
| LLaVA-v1.5-7B | 7B | 31.34 | 37.13 | 19.20 | 24.20 | 14.82 | 11.74 | – | 15.50 | – |
| LLaVA-NeXT-Interleave | 7B | – | 51.30 | 32.40 | 34.00 | 17.99 | 13.81 | – | 14.70 | – |
| LLaVA-OneVision | 7B | 64.20 | 48.20 | 40.32 | 63.20 | 26.20 | 18.30 | 13.69 | 18.80 | 36.61 |
| LLaVA-OneVision-VISC-150K | 7B | 66.36 | 50.24 | 46.52 | – | – | – | – | – | – |
| Migician-630K | 7B | 69.12 | 51.53 | 52.32 | 58.90 | 30.10 | 19.60 | 8.73 | 14.88 | 38.15 |
| InternVL2 | 8B | 65.40 | 50.90 | 42.00 | 58.30 | 37.00 | 17.40 | 25.85 | – | – |
| InternVL2.5 | 8B | 67.00 | 54.80 | 46.70 | 64.40 | 39.50 | 19.70 | 30.92 | 16.84 | 42.48 |
| Deepseek-VL-7B | 7B | – | 40.90 | 24.64 | 37.20 | 18.42 | 13.82 | – | 14.50 | – |
| Qwen2-VL-7B | 7B | 69.60 | 53.20 | 52.76 | 58.20 | 32.50 | 16.30 | 28.62 | 14.53 | 40.71 |
| Qwen2.5-VL-7B | 7B | 70.80 | 55.23 | 52.00 | 68.20 | 46.30 | 25.07 | 36.38 | 20.41 | 47.90 |
| *Multimodal Reasoning Models* | | | | | | | | | | |
| R1-VL-7B-260K$^*$ | 7B | 64.81 | 55.00 | 46.58 | 63.50 | 40.00 | 24.70 | 24.31 | 18.77 | 42.62 |
| Qwen2-VL-7B-VISC-150K$^*$ | 7B | 69.12 | 55.34 | 52.76 | 58.50 | – | – | – | – | – |
| R1-Onevision-7B$^*$ | 7B | 71.76 | 32.38 | 46.48 | 64.10 | 46.40 | **29.90** | 36.96 | 19.06 | 43.38 |
| MM-Eureka$^*$ | 7B | 69.91 | 51.49 | 51.67 | **73.00** | 50.30 | 26.90 | 43.31 | 24.39 | 49.49 |
| Noisy-K12$^*$ | 7B | 73.15 | 54.23 | 52.13 | 72.90 | **52.80** | 28.90 | 41.27 | 24.99 | 50.05 |
| **Qwen-PeRL (Ours)**$^*$ | 7B | **76.39** | **58.53** | **54.23** | **73.00** | 49.56 | 28.26 | **43.38** | **25.68** | **51.13** |
| Δ *over the Baseline Model* | – | +5.59 | +3.30 | +2.23 | +4.80 | +3.26 | +3.19 | +7.00 | +5.27 | +3.23 |

R1-VL-7B-260K [58] and R1-OneVision-7B [55] are trained on 260K and 155K supervised samples before RL. MM-Eureka [34] and Noisy-K12 [30] are trained only with RL on K12 single-image math data. Our method follows the same RL-only setting, but focuses on multi-image scenario.

**SOTA Results**   As shown in Table 1, PeRL demonstrates exceptional performance across diverse benchmarks, achieving state-of-the-art results on multimodal natural scene understanding while maintaining competitive performance on mathematical reasoning tasks.

Our model achieves superior results with 76.39 on Mantis-Eval and 58.53 on BLINK, substantially outperforming previous best models. This achievement is particularly significant considering our approach utilizes substantially less training data compared to traditional supervised fine-tuning methods. On single-image mathematical reasoning tasks, PeRL demonstrates competitive performance with scores of 73.00 on MathVista and 49.56 on MathVerse, comparable to specialized state-of-the-art models MM-Eureka and Noisy-K12.

Most notably, our model exhibits robust generalization on multi-image mathematical reasoning tasks despite these benchmarks being out-of-distribution relative to our training paradigm. This generalization capability is particularly valuable as our model was only trained on single-image mathematical reasoning and multi-image natural scene understanding datasets.

The comprehensive evaluation across 8 metrics yields an average score of 51.13, highlighting the effectiveness of our proposed training framework tailored specifically for interleaved multimodal tasks.

## 5.2   Ablation Study

**Permutation vs. Naive GRPO**   We investigate the effect of diversified permutation on training dynamics and generalization. As shown in Fig. 5a, Permutation GRPO gradually surpasses naive GRPO with more stable training. This is because the permutation disrupts fixed visual token positions, exposing the model to harder cases initially, but enabling it to overcome positional bias through adaptation. To further clarify the principle behind our method, we include a concise theoretical justification. Detailed derivations and formal proofs are provided in Appendix.

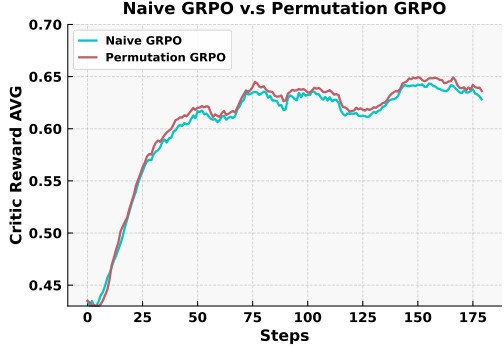
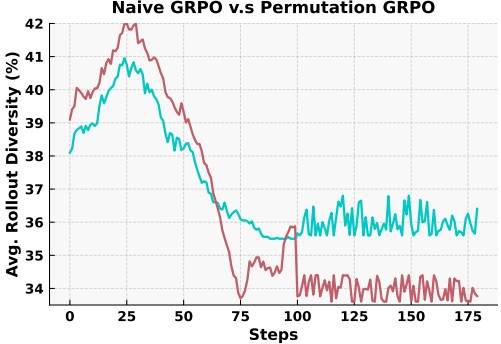

(a) Reward curve comparison.                (b) Rollout diversity comparison.

Figure 5: Permutation GRPO achieves more consistent training behavior.

Table 2: Permutation Ablation: $n_s$ is the number of permuted samples per original input. The total number of rollouts per input 12 is held constant. $n_s = 0$ indicates no permutation applied, while $n_s = 2$ indicates two permuted samples per input.

| Permutation | Mantis-Eval | BLINK | MMIU | MathVista | MathVerse | MathVision | Remi | MV-MATH |
|---|---|---|---|---|---|---|---|---|
| $n_s = 0$ | 75.93 | 56.91 | 52.26 | 71.8 | 49.20 | 26.51 | 39.62 | 24.02 |
| $n_s = 1$ | 76.39 | 58.53 | 54.23 | 73.0 | 49.56 | 28.26 | 41.65 | 25.68 |
| $n_s = 2$ | 77.78 | 59.50 | 54.55 | 72.2 | 50.38 | 27.89 | 39.04 | 24.54 |
| $n_s = 3$ | 74.07 | 57.02 | 52.17 | 70.9 | 46.39 | 27.00 | 38.62 | 23.25 |

Table 3: Single v.s. Multiple Image Disentanglement Ablation

| Data Setting | | Benchmark | | | | | | | |
|---|---|---|---|---|---|---|---|---|---|
| Multi-image | Single-image | Mantis-Eval | BLINK | MMIU | MathVista | MathVerse | MathVision | Remi | MV-MATH |
| ✗ | ✗ | 70.80 | 55.23 | 52.00 | 68.20 | 46.30 | 25.07 | 36.38 | 20.41 |
| ✓ | ✗ | 74.54 | 57.84 | 53.70 | 68.5 | 46.1 | 26.60 | 39.12 | 20.58 |
| ✗ | ✓ | 69.23 | 53.24 | 51.67 | 71.9 | 51.4 | 27.20 | 41.67 | 24.26 |
| ✓ | ✓ | 76.39 | 58.53 | 54.23 | 73.0 | 49.56 | 28.26 | 41.65 | 25.68 |

To evaluate the impact on output diversity, we compute the average intra-batch cosine similarity of rollout embeddings using the BGE-Large model[3]. As shown in Fig. 5b, Permutation GRPO maintains higher diversity throughout training. Given the same prompt, the policy sees varied visual token orders, which encourages learning order-invariant representations and improves robustness.

**Permutation intensity** We investigate the effect of the number of augmented permutations $n_s$. As shown in Table 2, increasing $n_s$ from 0 (i.e., the naive GRPO baseline without permutation) to 2 improves performance on most benchmarks. For instance, Mantis-Eval increases from 75.93 to 77.78, and BLINK from 56.91 to 59.50, indicating that exposure to varied input orders promotes positional invariance. The improvements are particularly notable on multi-image tasks with complex spatial structures, such as BLINK and MathVerse. However, $n_s = 2$ yields marginal degradation on reasoning-centric benchmarks (e.g., MathVista and Remi) and incurs additional training cost due to reduced key-value cache reuse. These findings suggest a trade-off: permutation augmentation enhances spatial generalization but may impair reasoning performance and training efficiency.

**Single vs. Multi-Image** Table 3 shows the impact of single-image and multi-image training data. Training exclusively on single-image datasets leads to limited generalization on multi-image benchmarks (e.g., significant drops on Mantis-Eval and BLINK), while training solely on multi-image data similarly reduces effectiveness on single-image reasoning benchmarks (e.g., MathVista). Interestingly, even without explicitly training on multi-image math reasoning data, performance improves on out-of-distribution benchmarks (Remi and MV-MATH), highlighting the model's implicit learning of both inter-image relationships and reasoning capabilities. Moreover, MathVision benefits from multi-image training, likely due to its composite nature of multiple concatenated math reasoning images, which aligns closely with learned multi-image reasoning patterns.

[3] https://huggingface.co/BAAI/bge-large-en-v1.5

## 6  Conclusion

In this paper, we investigate the multimodal reasoning capabilities of VLMs on general interleaved multi-image tasks. To address the challenges posed by spatial and positional reasoning, we introduce a multi-stage strategy that diversifies image order and resamples the training data distribution, enabling more effective and efficient learning. Experimental results demonstrate that our resulting model, Qwen-PeRL not only achieves superior performance on multi-image benchmarks but also generalizes well to single-image tasks. As future work, we aim to investigate adaptive permutation strategies that can dynamically adjust image order based on task difficulty or model uncertainty.

## 7  Acknowledgement

This work was supported by the National Key Research and Development Program of China (No. 2024YFB2808903).

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

# A  Technical Appendices and Supplementary Material

## A.1  Training and Evaluation Details

Table 4: Details of evaluation benchmarks.

| Benchmark | Description | #samples |
|-----------|-------------|----------|
| Mantis-eval | Multi-image General Understanding QA | 217 |
| BLINK | Multi-image General Understanding QA | 1901 |
| MMIU | Multi-image General Understanding QA | 11698 |
| MathVista | Single-image Math Reasoning QA | 1000 (testmini) |
| MathVerse | Single-image Math Reasoning QA | 3940 |
| MathVision | Single-image Math Reasoning QA | 3040 |
| Remi | Multi-image General Reasoning | 2600 |
| MV-Math | Multi-image Math Reasoning | 2009 |

We train our model on 8×H100 GPUs using the GRPO-based framework. The training batch size is 128, with a micro-batch size of 4 per GPU. The maximum prompt and response lengths are 8192 and 2048 tokens, respectively. We adopt Qwen2.5-VL-7B-Instruct[4] as the backbone, enabling gradient checkpointing and FSDP without offloading. KL loss is applied with a coefficient of 0.01, while entropy regularization is disabled. The model is optimized for 2 epochs with a learning rate of $1 \times 10^{-6}$.

For evaluation, we use greedy decoding with temperature set to 0, top-p to 1, top-k to -1, and a maximum generation length of 2048. Evaluation benchmarks include MathVista, MathVerse, MathVision, and BLINK (configured via VLMEvalKit), while Mantis-Eval, MMIU, and MV-MATH (evaluated with official code via vLLM). All evaluations follow consistent decoding settings.Details are shown as Table 4 and prompt A.3

## A.2  Case study

As shown in Fig 6, we conduct a detailed comparison of the response quality across different types of models, including the proprietary model o4-mini-high, open-source base models, and previously trained interleaved models. We observe that the base models exhibit various types of errors, which can be broadly categorized into several patterns: missing relevant images, misinterpreting images, producing incorrect answers despite correct reasoning, and reaching incorrect conclusions despite accurate observations of details. For instance, in the beetle question shown above, Qwen frequently makes errors where the reasoning is correct but the final answer is wrong, or where some relevant visual information is overlooked. These are the types of failures we aim to address.

Besides, in case fig 8, we observe a consistent pattern in model behavior across the permutations of images: the model exhibits a stronger alignment between the reference image and the image presented first in the sequence of options. This suggests current VLMs have a positional bias in interleaved vision tasks.

---

[4] https://huggingface.co/Qwen/Qwen2.5-VL-7B-Instruct

Question: How many beetles are climbing on the balls in the image?
A. Four B. Five C. None of the choices provided D. Two E. One

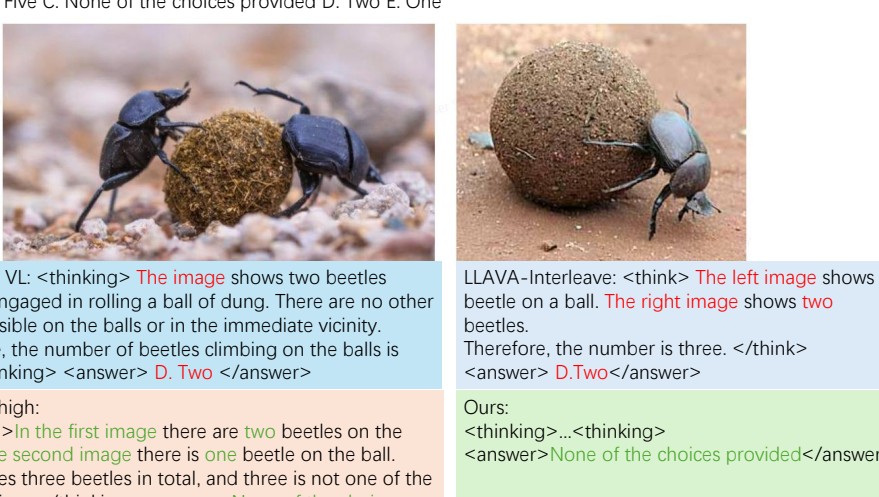

Qwen 2.5 VL: <thinking> The image shows two beetles actively engaged in rolling a ball of dung. There are no other beetles visible on the balls or in the immediate vicinity. Therefore, the number of beetles climbing on the balls is two.</thinking> <answer> D. Two </answer>

LLAVA-Interleave: <think> The left image shows a beetle on a ball. The right image shows two beetles. Therefore, the number is three. </think> <answer> D.Two</answer>

o4-mini-high:
<thinking>In the first image there are two beetles on the ball. In the second image there is one beetle on the ball. That makes three beetles in total, and three is not one of the listed options.</thinking> <answer>None of the choices provided</answer>

Ours:
<thinking>...<thinking>
<answer>None of the choices provided</answer>

Figure 6: A showcase for motivation

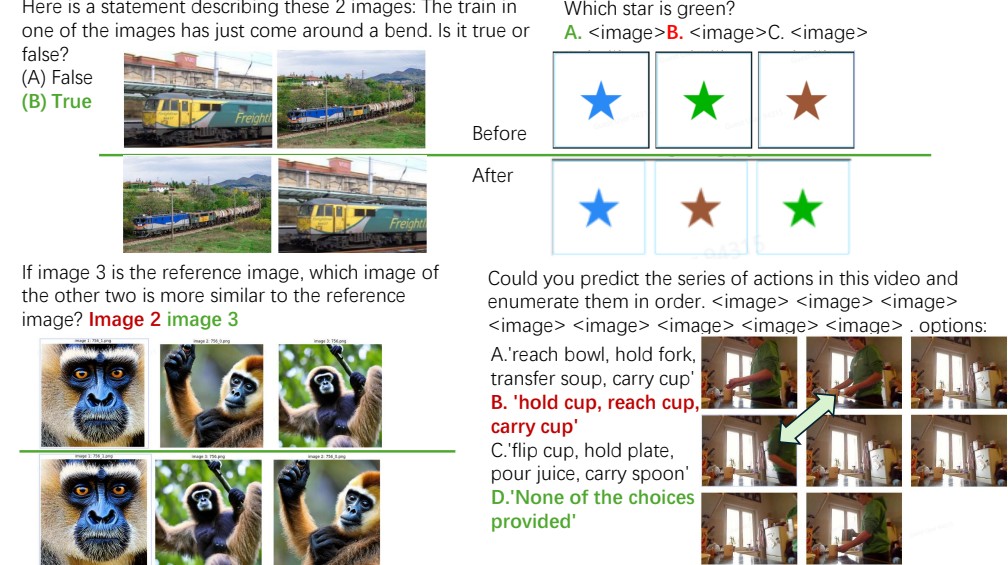

Figure 7: Training Data preprocessing pipeline explanation and several cases

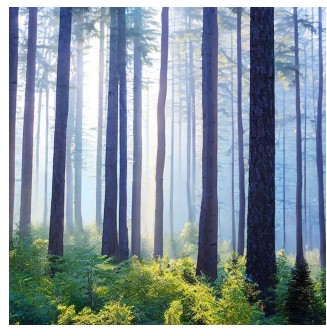 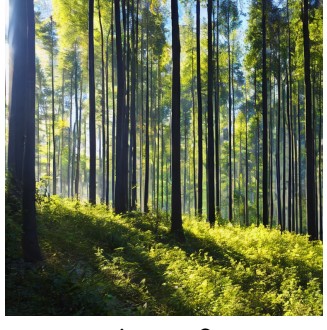 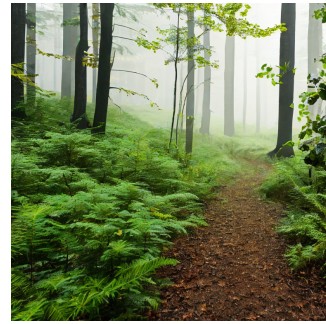

| <image_1> | <image_2> | <image_3> |

## Before Permuation

Which image is more similar to the reference image <image_1>? Options: A. <image_2> B. <image_3>

**Qwen-2.5-VL-7B**: To determine which image is more similar to the reference image, let's analyze the key elements of both images:

- Reference Image: 1) Features tall trees with a dense forest setting. 2) The lighting appears to be natural, possibly early morning or late afternoon, with sunlight filtering through the trees.
- Option A: 1) Also shows a forest scene with tall trees. 2) There is less mist or fog compared to the reference image.
- Option B: 1) Displays a forest path surrounded by lush greenery. 2) There is no mist or fog present, and the overall atmosphere is more open and less dense than the reference image.

Comparing these elements, Option A is closer to the reference image because it maintains the tall trees and dense forest setting. ✓

## After Permuation

Which image is more similar to the reference image <image_1>? Options: A. <image_3> B. <image_2>

**Qwen-2.5-VL-7B**: To determine which image is more similar to the reference image, let's analyze the key elements of both images:

- Reference Image: 1) Features tall trees with a dense forest setting. 2) The lighting suggests early morning or late afternoon, with sunlight filtering through the trees. 3) The ground is covered with green foliage, and there is a misty atmosphere.
- Option A: 1) Also features a forest with tall trees. 2) There is a path visible, which is not present in the reference image. 3) The lighting appears to be diffused, possibly due to the mist, but it is less pronounced than in the reference image. 4) The ground is covered with ferns and other greenery, similar to the reference image.
- Option B: 1) Features tall trees with a dense forest setting. 2) The lighting is bright and direct, suggesting midday sun. 3) The ground is covered with green foliage, but the overall scene lacks the misty atmosphere seen in the reference image.

Given this analysis, option B lacks the misty atmosphere and has brighter lighting, making it less similar to the reference image. The image that is most similar to the Reference Image is Option A. ✗

Figure 8: Case study demonstrating positional bias. The model correctly identifies the similar image when presented first (Before Permutation) but fails when the order is swapped (After Permutation).

### A.3 prompt

#### A.3.1 Prompt for data preprocessing

During the data preprocessing stage, we employed GPT-4o in conjunction with the prompts provided below to reformat questions and perform semantic variation checks.

---

## Format Rephrase

Your task is to convert a given QA conversation into a multiple-choice question format, and determine whether the question follows the given question type.

### Question Type Definition

1) A question is considered a **Reference-Image Comparison** if it satisfies all of the following conditions:

- The question presents three or more images (e.g., "`<image> <image> <image>`").
- One image is clearly identified as the **reference image** (e.g., "image 1 is the reference").
- The question asks which of the remaining images is most similar to or most different from the reference image.
- The answer options correspond only to the non-reference images.

2) ...

### Task Instructions

**1) Convert the original QA pair into a multiple-choice question:**

- Rephrase the assistant's response into an answer option (e.g., "A", "B", "C").
- Use placeholder tokens (`<image>`) in both the question and the options.
- Include only the images being compared (exclude the reference image from the options).
- Format the question strictly according to the example below.

**2) Determine the type of the question.**

### Output Format

Return a JSON object with the following structure:

```
{
  "question": "<multiple-choice question in specified format>",
  "answer": "<correct option letter>",
  "question_type": "<ReferenceComparison or Other>"
}
```

### Example

**Input:** "Question: Answer the following question: Here are three images: `<image_1>` `<image_2>` `<image_3>`. If image 1 is the reference image, which image of the other two is more similar to the reference image? Answer: The image that is more similar to the reference image is image 2."

**Output:**

```
{
    "question": "Answer the following question: Which image
        is more similar to the reference image <image_1>?
        Options: A. <image_2> B. <image_3>",
    "answer": "A",
    "question_type": "ReferenceComparison"
}
```

---

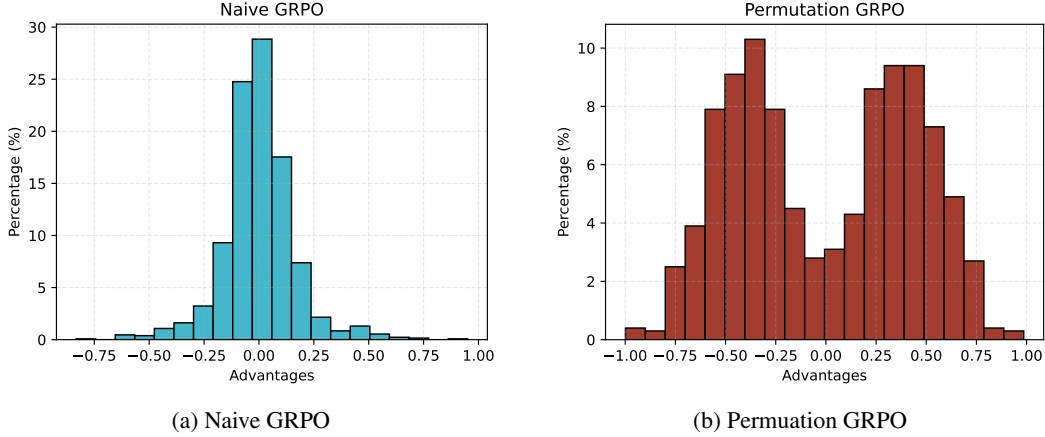

|(a) Naive GRPO|(b) Permuation GRPO|

Figure 9: Difference on advantage between Naive GRPO and Permuation GRPO during training stage.

---

**Semantic Variation Check**

You will be given a question involving one or more images, which are represented using image tokens. The token `<image>` represents an actual image. The tokens `<image_1>`, `<image_2>`, etc., refer to specific images by their positions (e.g., `<image_1>` refers to the first image, `<image_2>` to the second image).

**Your task is to determine the following:**

1. If only the order of the images (e.g., `<image_1>`, `<image_2>`, etc.) is changed, would the answer to the question need to change?
2. Is the question structured such that a single main image appears in the question body, and other images are referenced in the choices?

**Respond in the following JSON format:**

```
{
    "should_change": true or false,
    "is_multichoice_images": true or false
}
```

*Note: This applies to both multiple-choice and fill-in-the-blank questions involving image references.*

---

### A.3.2 Prompt for training and evaluation

**Reasoning Format**

**Instruction:**
You first think about the reasoning process as an internal monologue and then provide the final answer.
The reasoning process must be enclosed within `<think> </think>` tags.
The final answer must be put in `\boxed{}`.

---

### A.4 Adavantage Differences

As Fig 9 shows,the distinction becomes clearer when examining the advantage distributions during training. For multi-image inputs, Naive GRPO yields a distribution sharply peaked around zero, indicating that a large portion of training examples contribute negligible or ineffective gradient signals.

In contrast, the permutation-based GRPO introduces greater input diversity by altering image order, which encourages the model to genuinely capture positional biases. This diversification leads to more informative advantage signals, facilitating more effective gradient updates during optimization.

## A.5 Theoretical Analysis

Here's a theoretical analysis of why PeRL is better than GRPO in reducing positional bias. Following the notation, we start by defining GRPO's optimization problem:

$$\max_{\theta} \ \mathbb{E}_{x \sim \rho} \ \mathbb{E}_{o \sim \pi_{\text{old}}(\cdot | q)} \ f_{\epsilon}\Big(\frac{\pi_{\theta}(o \mid x)}{\pi_{\text{old}}(o \mid x)}, \ A(x, o)\Big) \ - \ \beta \, \mathrm{KL}\big(\pi_{\theta} \, \| \, \pi_{\text{ref}}\big),$$

where the advantage is defined as

$$A(x, o) = \frac{r(x, o) \ - \ \mathbb{E}_{o' \sim \pi_{\text{old}}(\cdot | x)}\big[r(x, o')\big]}{\sqrt{\mathrm{Var}_{o' \sim \pi_{\text{old}}(\cdot | x)}\big[r(x, o')\big] + \epsilon}}.$$

Recall that our reward is a verifiable reward that evaluates the correctness of a reasoning or the execution of the code, meaning that

$$r(x, o) \in \{0, 1\}.$$

We note the probability of success $p$ of the old policy:

$$p := p_{\theta_{\text{old}}}(x) = \mathbb{P}_{o \sim \pi_{\text{old}}(\cdot | x)}\big(r(x, o) = 1\big).$$

Hence, we have for the mean and variance of a Bernoulli random variable:

$$\mathbb{E}_{o' \sim \pi_{\text{old}}(\cdot | x)} r(x, o') = p, \qquad \mathrm{Var}_{o' \sim \pi_{\text{old}}(\cdot | x)} r(x, o') = p(1 - p).$$

This results in the following advantage:

$$A(x, o) = \begin{cases} \omega_{\varepsilon}^{+}(p) = \dfrac{1 - p}{\sqrt{p(1 - p)} + \varepsilon}, & \text{if } r(x, o) = 1, \\ \omega_{\varepsilon}^{-}(p) = \dfrac{p}{\sqrt{p(1 - p)} + \varepsilon}, & \text{if } r(x, o) = 0. \end{cases}$$

As established by Theorems 1 and 2 in [1], we obtain:

$$\pi_n(o \mid x) = \frac{1}{Z_{n-1}(x)} \, \pi_{\text{ref}}(o \mid x) \exp\left(\frac{1}{\beta}\Big[\omega_{\varepsilon}^{+}\big(p_{n-1}(x)\big)\mathbf{1}_{\{r(x,o)=1\}} - \omega_{\varepsilon}^{-}\big(p_{n-1}(x)\big)\mathbf{1}_{\{r(x,o)=0\}}\Big]\right),$$

where

$$Z_{n-1}(x) = p_{\text{ref}}(x) \exp\left(\frac{1}{\beta}\omega_{\varepsilon}^{+}\big(p_{n-1}(x)\big)\right) + \big(1 - p_{\text{ref}}(x)\big) \exp\left(-\frac{1}{\beta}\omega_{\varepsilon}^{-}\big(p_{n-1}(x)\big)\right).$$

Define

$$h_{\varepsilon, p_{\text{ref}}}(p) = \frac{1}{1 + \frac{1 - p_{\text{ref}}}{p_{\text{ref}}} \exp\left(-\frac{1}{\beta} \frac{1}{\sqrt{p(1-p)+\varepsilon}}\right)}.$$

GRPO evolves as

$$p_n^{\text{GRPO}}(x) = h_{\varepsilon, p_{\text{ref}}(x)}\big(p_{n-1}^{\text{GRPO}}(x)\big).$$

Similar to Theorems 1 and 2, we can show that PeRL updates as

$$p_n^{\text{PeRL}}(x) = h_{\varepsilon, p_{\text{ref}}(x)}\big(p_{n-1}^{\overline{\text{PeRL}}}(x)\big),$$

where $p_{n-1}^{\overline{\text{PeRL}}}(x)$ denotes the average accuracy of all permutations of the original input $x$.

—

**Proof that PeRL has less positional bias than GRPO.**   For $n = 0$, we have

$$p_0^{\text{GRPO}}(x_{\min}) \le p_0^{\text{PeRL}}(x_{\min}) \le p_0^{\text{PeRL}}(x_{\max}) \le p_0^{\text{GRPO}}(x_{\max}),$$

which holds because all models start from the same reference policy. Here, $p(x_{\min})$ and $p(x_{\max})$ represent the minimum and maximum accuracies across all image permutations, and we assume these correspond to the same permutations for all methods and do not change during training.

For iteration $n - 1$, assume

$$p_{n-1}^{\text{GRPO}}(x_{\min}) \le p_{n-1}^{\text{PeRL}}(x_{\min}) \le p_{n-1}^{\text{PeRL}}(x_{\max}) \le p_{n-1}^{\text{GRPO}}(x_{\max}).$$

Since $h$ is increasing for $p \in [1/2, 1]$, which holds for most of our training data, we obtain:

$$p_n^{\text{PeRL}}(x_{\max}) = h_{\varepsilon, p_{\text{ref}}(x_{\max})}\big(p_{n-1}^{\overline{\text{PeRL}}}(x)\big) \le h_{\varepsilon, p_{\text{ref}}(x_{\max})}\big(p_{n-1}^{\text{PeRL}}(x_{\max})\big) \le h_{\varepsilon, p_{\text{ref}}(x_{\max})}\big(p_{n-1}^{\text{GRPO}}(x_{\max})\big) = p_n^{\text{GRPO}}(x_{\max}).$$

Similarly,

$$p_n^{\text{PeRL}}(x_{\min}) \ge p_n^{\text{GRPO}}(x_{\min}).$$

Thus,

$$p_n^{\text{GRPO}}(x_{\min}) \le p_n^{\text{PeRL}}(x_{\min}) \le p_n^{\text{PeRL}}(x_{\max}) \le p_n^{\text{GRPO}}(x_{\max}).$$

By induction, this inequality holds for any step $n$ under the assumption. This shows that PeRL policy's success probabilities are less sensitive to input permutations than GRPO's, proving its effectiveness in reducing positional bias.

**Enhanced Generalization via Invariance.**   By forcing the policy to be robust to permutations, we implicitly guide the model to learn an order-invariant representation. This means the model must base its decisions on the semantic content of the images themselves, rather than on superficial positional cues. Learning invariant representations is a classic and powerful principle for improving generalization, as it ensures the model performs robustly on unseen data where permutations may differ from the training set.

**Improved Training Stability.**   Permutation also unintentionally increases the effective difficulty for any given input $x$. This lowers the probability of the model answering all inputs correctly, which in turn prevents the advantage estimate from collapsing toward zero and causing ineffective gradient updates—an issue common with overly simple samples. This process ensures a more consistent advantage signal for a single sample across its permutations (i.e., it reduces intra-group variance), which fundamentally stabilizes the overall training process.

