# OpenReview forum: "PeRL: Permutation-Enhanced Reinforcement Learning for Interleaved Vision-Language Reasoning"
_NeurIPS.cc/2025/Conference — NeurIPS 2025 poster_

### Official Review · Reviewer_rHiB · 2025-06-12

**Clarity:** 4
**Significance:** 3
**Originality:** 3
**Rating:** 4
**Confidence:** 4

**Summary:**

The paper proposes a reinforcement learning-based multimodal reasoning framework, PeRL, targeting the challenge of positional reasoning in interleaved vision-language inputs. By introducing image sequence permutation augmentation and a rollout resampling mechanism, the authors effectively alleviate the positional bias and training inefficiencies that current vision-language models face in multi-image tasks. The experimental evaluation spans multiple mainstream benchmarks, and the model achieves state-of-the-art performance on both multi-image natural scene understanding tasks.

**Questions:**

1. Could the authors provide a clearer justification of the novelty of the proposed approach?

2. Could more theoretical analysis be included? I encourage the authors to elaborate on the theoretical contribution boundaries of their method—particularly how the permutation mechanism impacts generalization, and whether it can be shown to improve training stability in a principled way.

3. How can the accuracy of the permutation-based samples generated by GPT-4o be ensured?

**Ethical Concerns:**

["NO or VERY MINOR ethics concerns only"]

**Final Justification:**

We have carefully reviewed the authors' rebuttal and the comments from other reviewers. While we appreciate the authors' responses and additional clarifications, several concerns remain unresolved. Notably, multiple reviewers, including myself, raised concerns regarding the accuracy and reliability of the permutation-based samples generated by GPT-4o. Furthermore, specific components of the method—such as the rule-based filtering in preprocessing and the design choices around model selection (e.g., using Qwen2.5-VL-7B for rollout filtering and GPT-4o for modification judgment)—lack sufficient justification. We encourage the authors to carefully revise the paper in light of all reviewer comments to better articulate the theoretical foundations, clarify the data generation process, and strengthen the empirical analysis. I'm looking forward to the final version of this paper. I will increase my score to "boardline acceptance" level.

**Limitations:**

Yes.

**Paper Formatting Concerns:**

None.

**Quality:**

3

**Strengths And Weaknesses:**

In my understanding, the paper has two core contributions: (1) A systematic preprocessing pipeline and semantic consistency check applied to the original multi-image dataset, which reduces and balances the QA difficulty distribution prior to training, significantly improving the stability and convergence of the reinforcement learning process; (2) An enhancement to the GRPO policy optimization algorithm, which leverages the task’s permutation property to increase the number of samples used for advantage estimation—combining responses from standard GRPO sampling and permutation-based augmentation—potentially leading to a more accurate average advantage baseline and thus better policy updates.

The paper has the following strengths: comprehensive experimental design across diverse evaluation settings, clear and well-structured presentation, and method development that reflects solid research thinking and practical value.

However, there are also several weaknesses:
1. The technical contributions are relatively straightforward, focusing primarily on data and training process optimization. The RL customization for this specific multimodal positional reasoning task feels insufficiently explored or underdeveloped.
2. The paper lacks in-depth theoretical analysis or quantitative justification regarding generalization or error control, making it appear somewhat empirical.

---

> ### Author Rebuttal · Authors · 2025-07-31
>
> We sincerely thank you for your insightful review and accurate summary of our work. We are greatly encouraged by your recognition of our two core contributions—the data pipeline and GRPO enhancement—and your positive assessment of our comprehensive experiments, clear presentation, and practical value. We address your specific points below.
>
> ---
>
> ### **W1&Q1: The technical contributions are relatively straightforward, focusing primarily on data and training process optimization. The RL customization for this specific multimodal positional reasoning task feels insufficiently explored or underdeveloped. Could the authors provide a clearer justification of the novelty of the proposed approach?**
>
> Thank you for this critical feedback. We agree that this is a nascent area of research, and we respectfully frame our work as a foundation, rather than an exhaustive exploration of using RL for this specific challenge.
>
> To our knowledge, this is the **first attempt to use RL to directly address positional bias** in VLMs. Our customization was deliberately designed to tackle two fundamental problems that arise when applying RL to multi-image tasks:
>
> 1. **On the model's reasoning ability:** Positional bias causes critical failures in understanding inter-image relationships. For instance, a model might generate a caption with incorrect positional cues (e.g., describing the second image as the first), which misleads its own subsequent reasoning steps. By enforcing order-invariance, our RL approach improves the model's core ability to locate the correct evidence regardless of image sequence, reducing these types of pattern-matching errors.
>
> 2. **On the stability of RL training:** A model sensitive to permutation poses a significant challenge for policy optimization. The same logical reasoning might receive a high reward in one image order and a low reward in another, leading to noisy and biased advantage estimates. Our method of averaging advantages across permutations directly stabilizes the training signal, ensuring the policy update is based on the actual reasoning quality, not on the arbitrary order of inputs.
>
> Therefore, while the technical implementation may seem straightforward, it is a principled solution designed to make the application of RL in this context both feasible and effective. We believe this work opens up a new and promising direction for more sophisticated RL-based solutions in the future.
>
> ---
>
> ### **W2&Q2: Could more theoretical analysis be included? I encourage the authors to elaborate on the theoretical contribution boundaries of their method—particularly how the permutation mechanism impacts generalization, and whether it can be shown to improve training stability in a principled way.**
>
>
> We sincerely thank the reviewer for this thoughtful suggestion. We have conducted a deeper investigation into the theoretical underpinnings of our method.
>
> **Here's a theoretical analysis of why PeRL is better than GRPO in reducing positional bias.**
>
> Following the notation in [1], we start by defining GRPO’s optimization problem:
>
> $$\max_{\theta}\;
> \mathbb{E}\_{x\sim \rho}
> \\mathbb{E}\_{o\sim \pi\_{\mathrm{old}}(\cdot\mid q)}
> f\_{\epsilon}\Bigl(
> \frac{\pi\_{\theta}(o\mid x)}{\pi\_{\mathrm{old}}(o\mid x)}
> \,A(x,o)
> \Bigr)
> \-\\beta\\mathrm{KL}\bigl(\pi\_{\theta}\|\pi\_{\mathrm{ref}}\bigr),
> $$
> where the advantage is:
>
> $$
> A(x,o)=\\frac{\\,r(x,o)\\;-\\;\\mathbb{E}\_{o'\\sim \\pi\_{\\mathrm{old}}(\\cdot\\mid x)}\\bigl[r(x,o')\\bigr]\\;}
> {\\sqrt{\\mathrm{Var}\_{o'\\sim \\pi\_{\\mathrm{old}}(\\cdot\\mid x)}\\bigl[r(x,o')\\bigr]+\\epsilon}\\,}
> $$
>
> Recall that our reward is a verifiable reward that evaluates the correctness of a reasoning or the execution of the code, meaning that:
>
> $$ r(x,o) \\in \\{0,1\\} $$
>
>
> We note the probability of success \( p \) of the old policy:
>
> $$ p := p\_{\\theta_{\\text{old}}}(x) = \\mathbb{P}\_{o \\sim \\pi\_{\\text{old}}(\\cdot|x)}\\bigl(r(x,o) = 1\\bigr) $$
>
> Hence, we have for the mean and variance of a Bernoulli random variable:
>
> $$ \\mathbb{E}\_{o' \\sim \\pi\_{\\text{old}}(\\cdot|x)} r(x,o') = p \\qquad \\text{and} \\qquad \\mathrm{Var}\_{o' \\sim \\pi\_{\\text{old}}(\\cdot|x)} r(x,o') = p(1-p). $$
>
>
> And this results in the following advantage:
>
> $$
> A(x,o) =
> \\begin{cases}
> \\omega_{\\varepsilon}^{+}(p) = \\dfrac{1 - p}{\\sqrt{p(1-p)} + \\varepsilon}, & \\text{if } r(x,o) = 1, \\\\
> \\omega_{\\varepsilon}^{-}(p) = \\dfrac{p}{\\sqrt{p(1-p)} + \\varepsilon}, & \\text{if } r(x,o) = 0.
> \\end{cases}
> $$
>
> As established by Theorem 1, 2 in [1], we obtain:
>
> $$
> \\pi\_n(o \\mid x) = \\frac{1}{Z\_{n-1}(x)} \\, \\pi\_{\\text{ref}}(o \\mid x)
> \\exp \\left(
> \\frac{1}{\\beta} \\left[
> \\omega_\\varepsilon^{+}\\bigl(p\_{n-1}(x)\\bigr) \\mathbf{1}\_{\\{r(x,o)=1\\}} - \\omega\_\\varepsilon^{-}\\bigl(p\_{n-1}(x)\\bigr) \\mathbf{1}\_{\\{r(x,o)=0\\}}
> \\right]
> \\right)
> $$
>
> where
>
> $$
> Z\_{n-1}(x) = p\_{\\text{ref}}(x) \\exp \\left( \\frac{1}{\\beta} \\omega\_\\varepsilon^{+}\\bigl(p\_{n-1}(x)\\bigr) \\right) + \\left( 1 - p\_{\\text{ref}}(x) \\right) \\exp \\left( - \\frac{1}{\\beta} \\omega\_\\varepsilon^{-}\\bigl(p\_{n-1}(x)\\bigr) \\right)
> $$
>
> Define:
>
> $$
> h_{\\varepsilon,p_{\\mathrm{ref}}}(p)=\\frac{1}{1+\\frac{1-p_{\\mathrm{ref}}}{p_{\\mathrm{ref}}}\\exp\\left(-\\frac{1}{\\beta}\\frac{1}{\\sqrt{p(1-p)+\\varepsilon}}\\right)},
> $$
>
>
> GRPO evolves as:
>
> $$ p_n^{GRPO}(x) = h_{\varepsilon,p_{\mathrm{ref}}(x)}(p^{GRPO}_{n-1}(x)). $$
>
> Similar to Theorems 1 and 2 from [1], we can show that PeRL updates as:
>
> $$ p_n^{PeRL}(x) = h_{\varepsilon,p_{\mathrm{ref}}(x)}({\bar{p}}_{n-1}^{PeRL}(x)) $$
>
> where  ${{\bar{p}}_{n-1}^{PeRL}(x)}$ denotes the average accuracy of all permutations of the original input x.
>
> **Proof that PeRL has less positional bias than GRPO:**
>
> For $n=0$, we have:
>
> $$ p_0^{GRPO}(x_{min}) \leq p_0^{PeRL}(x_{min}) \leq p_0^{PeRL}(x_{max}) \leq p_0^{GRPO}(x_{max}), $$
>
> which holds because all models start from the same reference policy. Here, $p(x_{min})$ and $p(x_{max}) $ represent the minimum and maximum accuracies across all image permutations, and we assume these correspond to the same permutations for all methods and do not change during training process.
>
> For iteration $n−1$ , assume:
>
> $$ p_{n-1}^{GRPO}(x_{min}) \leq p_{n-1}^{PeRL}(x_{min}) \leq p_{n-1}^{PeRL}(x_{max}) \leq p_{n-1}^{GRPO}(x_{max}), $$
>
> Since $h$  is increasing for $ p \in [1/2, 1] $, which holds for most of our training data, we obtain:
>
> $$ p_n^{PeRL}(x_{max}) = h_{\varepsilon,p_{\mathrm{ref}}(x_{max})}({\bar{p}}\_{n-1}^{PeRL}(x)) \leq h_{\varepsilon,p_{\mathrm{ref}}(x_{max})}(p_{n-1}^{PeRL}(x_{max})) \leq h_{\varepsilon,p_{\mathrm{ref}}(x_{max})}(p_{n-1}^{GRPO}(x_{max})) = p_n^{GRPO}(x_{max}). $$
>
> Similarly:
>
> $$ p_n^{PeRL}(x_{min}) \geq p_n^{GRPO}(x_{min}). $$
>
> Thus:
>
> $$ p_n^{GRPO}(x_{min}) \leq p_n^{PeRL}(x_{min}) \leq p_n^{PeRL}(x_{max}) \leq p_n^{GRPO}(x_{max}). $$
>
> By induction, this inequality holds for any step n under the assumption. This shows that PeRL policy's success probabilities are less sensitive to input permutations than GRPO's, proving its effectiveness in reducing positional bias.
>
> ---
>
> **Enhanced Generalization via Invariance** By forcing the policy to be robust to permutations, we are implicitly guiding the model to learn an **order-invariant representation**. This means the model must base its decisions on the semantic content of the images themselves, rather than on superficial positional cues. Learning invariant representations is a classic and powerful principle for improving generalization, as it ensures the model performs robustly on unseen data where permutations may differ from the training set.
>
> **Improve training stability** Specifically, permutation unintentionally increases the effective difficulty for any given input x. This lowers the probability of the model answering all inputs correctly, which in turn prevents the advantage estimate from collapsing toward zero and causing ineffective gradient updates—an issue common with overly simple samples.
>
> This process ensures a more consistent advantage signal for a single sample across its permutations (i.e., **it reduces intra-group variance**), which fundamentally stabilizes the overall training process.
>
>
>
> ---
>
> ### **Q3: Quality Assurance for Permuted Samples Generated by GPT-4o**
>
> We adopted a two-step strategy to ensure correctness:
>
> 1. **Rule-Based Prompting:**  We designed a detailed prompt with explicit rules for how answers must change when permuting images. For instance, if all multiple-choice options are images, swapping them requires updating the correct answer label accordingly. We also included few-shot examples to guide GPT-4o. A partial prompt with key rules and examples is included in Appendix A.3.
>
> 2. **Manual Validation:**  To assess the accuracy of the generated permutations, we randomly sampled 200 items for manual validation, and found that **97.5%** of them were correctly generated, which demonstrates that our approach produces highly reliable permutation results in practice.
>
> This combination of carefully designed rule-based prompting and targeted manual validation ensured the overall accuracy of our generated data.
>
> We sincerely hope our reply can address your concern.
>
> ---
> ### **References**
>
>
> [1] Mroueh, Youssef. "Reinforcement Learning with Verifiable Rewards: GRPO's Effective Loss, Dynamics, and Success Amplification." arXiv preprint arXiv:2503.06639 (2025).

---

> > ### Comment · Reviewer_rHiB · 2025-08-01
> >
> > Thank you for your response. My concern has been fully addressed.

---

> > > ### Author Response · Authors · 2025-08-01
> > >
> > > Thank you for your careful reading! We would like to know if there is any additional experiment that we can conduct to make the work stronger.
> > > We are glad that our responses can address your concern, and we would kindly ask you to adjust your review score accordingly.

---

> > > > ### Comment · Reviewer_rHiB · 2025-08-01
> > > > **Update the score**
> > > >
> > > > We will carefully review the paper and the comments from other reviewers again, and then make a final decision based on the comprehensive consideration. If we have any further questions, we will discuss them with you again.

---

> > > > > ### Author Response · Authors · 2025-08-07
> > > > > **Re:Update the score**
> > > > >
> > > > > Dear Reviewer rHiB,
> > > > >
> > > > > With the review deadline approaching, we want to sincerely thank you again for your thoughtful feedback on the theoretical soundness of our method. Your comments have been invaluable in helping us improve the paper.
> > > > >
> > > > > During the rebuttal process, the other reviewers (e.g., 4xuh and P1AW) also raised constructive points and have expressed positive views about the robustness of our work. In response to the helpful feedback from all reviewers, we have added new results showing that our approach:
> > > > >
> > > > > - has a clear motivation and effectively reduces positional bias;
> > > > >
> > > > > - delivers consistent, strong generalization in more complex settings, directly supporting the theoretical points you raised;
> > > > >
> > > > > - includes an alternative, open-source, high-quality data-processing pipeline.
> > > > >
> > > > > We genuinely appreciate your insights and would welcome any further thoughts you might have as you finalize your decision. We’d be very happy to clarify or discuss further at any time. Thank you again for your effort and active engagement.
> > > > >
> > > > > The authors of Paper 15073

---

> > > > > > ### Comment · Reviewer_rHiB · 2025-08-08
> > > > > >
> > > > > > Don't worry, we gave a borderline accept score.

---

### Official Review · Reviewer_HKDU · 2025-06-28

**Clarity:** 3
**Significance:** 2
**Originality:** 2
**Rating:** 4
**Confidence:** 3

**Summary:**

This paper introduces a data permuation method to expand the input space for multiple-image positional reasoning. Correspondingly, a modified GRPO is proposed to take the augmented trajectories to compute advantages. The method has been tested on 5 commonly used multi-image benchmarks and 3 single-image benchmarks, showing superior performance over previous approaches.

**Questions:**

1 what's the motivation of rule-based filter as the first preprocessing step? how much data has been left after this first step filtering?

2 what's the basis or reasons to choose which model for which preprocessing step? In section 3.3, Qwen2.5-VL-7B is adopted for rollout filtering and GPT-4o for assessing whether the answer should be modified.

3 After premutation of the images, if the answer has to be adapted, what's the process to modify the answer? Is it automatically generated or human justtification needed?


4 Permutation GRPO requires more rollouts to compute the advantage. What's the computational cost comparing to naive GRPO ?

**Ethical Concerns:**

["NO or VERY MINOR ethics concerns only"]

**Final Justification:**

I have read the responses by the authors and the other reviews. The authors have cleared my computational cost concenrs compared against native GRPO and will add more details about data processing in their revision. I would like to keep my original rating.

**Limitations:**

Yes

**Quality:**

2

**Strengths And Weaknesses:**

Strengths:
1 The whole paper is well strcutred and layout is clear.

2 The method achives SOTA performance on commonly used benchmarks.

3 The proposed approach method is straightforward to understand.

Weakness:
1 Detailed contexts can be more clearly presented. In the data processing, the authors can elaborate the filtering process and how much data is left before and after filtering.

2 When the image has been permuted, if the answer has to be modified, it may have the scaling problem.

3  Permutation GRPO may take more rollouts compared against native GRPO.

---

> ### Author Rebuttal · Authors · 2025-07-31
>
> Thank you for your valuable and detailed feedback. We are motivated by your recognition of our work's **clear structure (S1)**, and **SOTA performance (S2)**. Below, we address the concerns as follows.
>
> ---
>
> ### **W1&Q1: Motivation of rule-based filtering and data statistics**
>
> Our work mainly focuses on scenarios involving multiple images. Therefore, this rule-based filter acts as a low-cost preprocessing step to filter samples that:
>
> - Contain only a single image or no images.
> - Cannot be verified by rule-based verifiers (e.g., Image caption or other general reasoning tasks)
>
> Starting from the **Mantis pre-training dataset [1] (721k samples)**, this step filters out over 80% of irrelevant data, leaving **137k samples** for downstream processing. We will include these statistics in the revised paper.
>
> ---
>
> ### **Q2: Rationale for model choices**
>
> We choose models based on two main considerations: **alignment with our RL training setup** and **the task complexity**.
>
> 1. **Qwen2.5-VL-7B for Rollout Filtering:**  We chose Qwen2.5-VL-7B because it is the base model we use for the subsequent training stage. For RL training to be effective, it is important to calibrate data difficulty to the model’s capacity. Using the base model itself for rollout filtering helps identify and remove samples that are too simple for it to learn from. After this step, the number of samples is further reduced from 137k to 22k.
>
> 2. **GPT-4o for Answer Adaptation:**  In contrast, the task of assessing and modifying answers after image permutation demands a higher level of abstract logical reasoning. This step focuses on how to accurately generate complex, high-quality training samples. We use GPT-4o for its state-of-the-art reasoning capabilities to perform this complex modification automatically, creating challenging data that is essential for our experiments.
>
> ---
>
> ### **W2&Q3: Process of adapting answers after image permutation and automatically generated**
>
> We appreciate the reviewer’s question regarding the scalability and reliability of this step. The adaptation process is fully automated, which makes it scalable:
>
> - We construct a detailed prompt, including explicit rules and few-shot examples, to guide GPT-4o in determining whether to modify answers after image permutation.
> - To ensure the quality of the automatically adapted answers, we manually inspected a representative subset and observed that they were generally accurate and consistent.
>
> We acknowledge that fully automated processes can introduce occasional errors; however, in practice, we found the overall quality to be sufficiently high for our experimental purposes. We will clarify this process and its limitations in the revised paper to provide a more balanced discussion.
>
> ---
>
> ### **Q3&W4: Permutation GRPO may take more rollouts compared against native GRPO  Permutation GRPO requires more rollouts to compute the advantage. What's the computational cost comparing to naive GRPO ?**
>
> Thank you for this important question regarding computational cost. We respectfully clarify that there may be a misunderstanding about the number of rollouts.
>
> As stated in the caption of **Table 2 in the original paper**, the total number of rollouts per input is **held constant at 12**, which is identical to the native GRPO setup. For example, when we use one permutation ($n_s=1$), we generate two sequences (the original and the permuted). Each sequence undergoes 6 rollouts, and the advantages from all 12 are then averaged. Therefore, our method does not require more rollouts.
>
> While your concern about the overall computational cost is reasonable. The overhead stems not from more rollouts, but from **reduced KV cache efficiency**. As $n_s$ increases, the frequent changes in image order alter the input context, forcing the model to switch or recompute the KV cache more often.
>
> To quantify this, we have performed a new analysis of the wall-clock training time per epoch for different $n_s$ values below, which we will add to the paper. This analysis confirms a modest and predictable increase in computational cost, which we believe is a reasonable trade-off for the significant multi-image performance gains shown in **Table 2**.
> | $ n_s$  | Training time per step/min |
> |------------|----------------------------|
> | $ n_s = 0 $ | 3.7                        |
> | $ n_s = 1 $| 5.0                        |
> | $ n_s = 2 $ | 9.2                        |
>
>
> ---
> ### **References**
>
> [1] Jiang, Dongfu, et al. *"Mantis: Interleaved Multi-Image Instruction Tuning."* Transactions on Machine Learning Research. (2024)

---

> > ### Author Response · Authors · 2025-08-05
> >
> > Dear Reviewer HKDU,
> >
> > I hope this message finds you well. As the discussion period is nearing its end with less than three days remaining, I want to ensure we have addressed all your concerns satisfactorily. If there are any additional points or feedback you'd like us to consider, please let us know. Your insights are invaluable to us, and we're eager to address any remaining issues to improve our work.
> >
> > Thank you for your time and effort in reviewing our paper.
> >
> > The authors of Paper 15073

---

### Official Review · Reviewer_P1AW · 2025-07-01

**Clarity:** 2
**Significance:** 3
**Originality:** 3
**Rating:** 5
**Confidence:** 2

**Summary:**

This work proposed PeRL to solve the hallucination issue when the images are shuffled in different order. This work discussed the problem that when shuffling the images, it gives the challenges for "spatial reasoning" and "position reasoning". By data augmentation (permuting image orders, rephrases text), the model can learn the invariants with different orders. Also this method applies the filtering mechanism for better training efficiency. Based on the evaluation results on multiple datasets, showing the higher performance than the SoTA baselines.

**Questions:**

N/A

**Ethical Concerns:**

["NO or VERY MINOR ethics concerns only"]

**Final Justification:**

The authors' response has addressed all my concerns. I change my rating to accept.

**Limitations:**

The limitation is discussed in the conclusion section.

**Quality:**

4

**Strengths And Weaknesses:**

Strength:

- The proposed method achieves the SoTA performance in a wide range of benchmarks, showing the effectiveness of the proposed method.
- Extensive and detailed ablations studies are conducted.
- The idea is clean, novel and effective.

Weakness:

- When calculating the reward, the rule-based evaluator is adopted (mentioned in Section 4). How's the rule-based evaluation works? Is the reward calculation accurately? As I know, if the reward calculation is noisy, the outcome of the training will be the issue. Is there any quantitative or qualitative analysis for this?
- The data preprocessing relies on the LLM (GPT4o)'s output (mentioned in Section 3.3). How do you ensure the high LLM's output quality? Also, relying on the close-sourced LLM will be a potential reproductive issue, especially when the model is deprecated by the provider (e.g. OpenAI). Can any open source LLMs achieves the similar quality for data preprocessing?
- This work focus on the multi-image reasoning. However, it only covers the images less than 5. Discussing the scenarios with much more images will be important. For example, evaluating on SlideVQA (each question is related to 20 slide images).

Minor Weakness:
- Typo: “Problem Formuation and Analysis” → “Problem Formulation and Analysis” (Section 3 title)

Overall, I think this work has no major issue, so I would like to give accept rating.

---

> ### Author Rebuttal · Authors · 2025-07-31
>
> Dear Reviewer P1AW,
>
> We sincerely thank you for your insightful and constructive feedback. We are greatly encouraged by your positive assessment, particularly that you found our core idea to be **clean, novel, and effective (S1)**. We also appreciate your recognition of our method's **state-of-the-art (SOTA) performance (S2)** across a wide range of benchmarks and your praise for our **extensive and detailed ablation studies (S3)**. Your feedback is invaluable for further improving our paper, and we address your specific questions below.
>
> ---
>
> ### **W1: How's the rule-based evaluation work? Is the reward calculation accurate? As I know, if the reward calculation is noisy, the outcome of the training will be the issue. Is there any quantitative or qualitative analysis for this?**
>
> In **GRPO**, the rule-based evaluation is straightforward because our tasks are framed as **multiple-choice questions**. The model generates a chain-of-thought (CoT) reasoning process and outputs its final prediction inside a clearly marked `<answer></answer>` tag. The reward calculation is **highly reliable** because it is based on an **exact match** between the model’s `<answer>` output and the **ground truth answer option**. This process is deterministic and not subjective:
>
> - If the predicted answer matches the correct option, the model receives a reward of **1**.
> - Otherwise, the reward is **0**.
>
> Because the reward is assigned through a **discrete and unambiguous multiple-choice format**, there is **no label noise** introduced by subjective judgment or ambiguous scoring criteria.
>
> This is unlike open-ended tasks, where partial credit or fuzzy matching may introduce inconsistencies when judged by GPT‑4o or rule-based matching.
>
> ---
>
> ### **W2: The data preprocessing relies on the LLM (GPT‑4o)'s output (mentioned in Section 3.3). How do you ensure the high LLM output quality? Also, relying on the closed-source LLM will be a potential reproducibility issue. Can any open-source LLMs achieve similar quality for data preprocessing?**
>
> #### **On the Quality of GPT‑4o Output for Data Preprocessing**
> We took several steps to ensure the **high quality and reliability** of the data preprocessing performed by GPT‑4o:
>
> - **Prompt Engineering**: We carefully designed prompts incorporating detailed rules and few-shot examples to guide the model's filtering decisions. For full transparency, the exact prompts are provided in **Appendix A.3**.
>
> - **Manual Verification**: To quantitatively validate the output, we manually inspected a random sample of 200 filtered data points. **Our audit revealed an accuracy rate of 97.5%**, suggesting a high degree of reliability in our preprocessing pipeline.
>
> #### **On Reproducibility and Closed-Source Models**
> We share the reviewer’s concern regarding reproducibility and explored an alternative **open-source pipeline**:
>
> - **Open-Source Alternative Experiment**: We used the **OmniCaptioner** [1] model to generate detailed captions for each image, replaced the images with placeholders (e.g., `<image_i>`), and utilized **Qwen2.5‑72B‑Instruct** language model to perform the same filtering task. This pipeline proved feasible—its accuracy was slightly lower than GPT‑4o’s but sufficient to demonstrate that **our method is reproducible with open-source models**.
>
> We chose GPT‑4o for our main experiments to guarantee the highest data quality, but we are confident that **future work can transition to fully open-source solutions** as their capabilities continue to improve.
>
> ---
>
> ### **W3: This work focuses on multi-image reasoning. However, it only covers images less than 5. Discussing scenarios with many more images will be important. For example, evaluating on SlideVQA (20 slide images per question).**
>
> Thank you for this insightful suggestion. Evaluating on scenarios with a much larger number of images is indeed crucial for demonstrating the scalability and generalizability of our method.
> First, we would like to clarify that the MMIU [2] benchmark, already included in our paper, has a wide distribution of image counts, with 36.4% of its samples containing more than 5 images. Following your advice, we have now included a more fine-grained breakdown of performance by image count in the last table, which shows PeRL consistently improves performance as the number of images increases.
> More importantly, inspired by your comment, we have conducted extensive new experiments on several challenging benchmarks that involve a large number of images. These include SlideVQA [3] (up to 20 images), DUDE [4] (up to 120+ images), and two video understanding tasks, MVBench [5] and TempCompass [6]. We consider video understanding an even more challenging form of multi-image reasoning, as it requires analyzing both semantic and relational cues across many frames.
>
>
> | Model          | SlideVQA ANLS | SlideVQA Exact Match | SlideVQA F1 Score | DUDE ANLS | MVBench ACC | TempCompass ACC |
> |----------------|---------------|----------------------|-------------------|-----------|-------------|-----------------|
> | Qwen-VL-2.5 7B | 56.9          | 43.9                 | 51.1              | 60.2      | 61.9        | 68.8            |
> | +PeRL          | **59.2 (+2.3)**   | **46.0 (+2.1)**          | **53.2 (+2.1)**       | **65.5 (+5.3)** | **64.5 (+2.6)** | **70.0 (+1.2)**     |
>
> As shown in above, PeRL demonstrates **consistent and significant performance gains** across these complex benchmarks.
>
> **Why does PeRL generalize so effectively?** To understand why PeRL generalizes so effectively, we conducted a deeper analysis on SlideVQA, breaking down the results by answer and reasoning types. We report two types of answer accuracy: (i) by answer type (Non-span, Single-span, Multi-span) and (ii) by reasoning type (single-hop, multi-hop).We found that PeRL's success is largely because the model truly learns to find and synthesize cross-span evidence from multiple slides. Notably, the performance gain on "Multi-span" questions (+3.6) is substantially higher than that from the GRPO baseline (-1.8). Furthermore, the model's "Multi-hop" reasoning ability also improves, benefiting from the generalization of the strong single-image reasoning capabilities PeRL maintains.
> This detailed analysis provides evidence that by mitigating positional bias, PeRL genuinely learns to understand the relationships between all images more holistically, thereby strengthening both its perception and reasoning capabilities.
>
> ---
> **(i) Answer type by answer-type**
>
> | Model            | Non-span   | Single-span | Multi-span |
> |------------------|-----------:|-----------:|-----------:|
> | Qwen-vl-2.5-7B   | 49.3       | 60.7       | 12.2       |
> | +GRPO            | 50.5 (+1.2) | 61.4 (+0.7) | 10.4 (−1.8) |
> | +PeRL            | **50.7 (+1.4)** | **62.9 (+2.2)** | **15.8 (+3.6)** |
>
> **(ii) Answer type by reasoning type**
>
> | Model            | single-hop   | multi-hop   |
> |------------------|------------:|-----------:|
> | Qwen-vl-2.5-7B   | 58.0        | 39.3       |
> | +GRPO            | **59.5 (+1.5)** | 39.8 (+0.5) |
> | +PeRL            | 59.3 (+1.3)  | **42.0 (+2.7)** |
>
>
>
>
> Finally, we further include more detailed **MMIU** results mentioned in original paper  below, which also show robust improvement. (The ‘6–10’ column indicates performance on the MMIU subset where the number of images per sample is greater than 6 and less than 10  )
>
> | Model          | 6–10       | 11–15       | 16–20       | 21–64       |
> |----------------|-----------|------------|------------|------------|
> | Qwen-vl-2.5 7B | 36.0      | 52.8       | 54.4       | 59.1       |
> | +PeRL          | **38.3 (+2.3)** | **56.0 (+3.2)** | **57.5 (+3.1)** | **60.6 (+1.5)** |
>
> - **Typo:** “Problem Formuation and Analysis” → “Problem Formulation and Analysis” (Section 3 title)
> Thank you for pointing this out. We will correct this typo in the revised version of the paper.
>
> We hope this response fully addresses your concerns. We are grateful for the constructive feedback, which has helped us strengthen our paper.
>
> ---
>
> ### **References**
>
> [1] Lu, Yiting, et al. *Omnicaptioner: One captioner to rule them all.* arXiv preprint arXiv:2504.07089 (2025).
>
> [2] Meng, Fanqing, et al. *MMIU: Multimodal Multi-image Understanding for Evaluating Large Vision-Language Models.* ICLR (2024).
>
> [3] Tanaka, Ryota, et al. *SlideVQA: A dataset for document visual question answering on multiple images.* AAAI (2023).
>
> [4] Van Landeghem, Jordy, et al. *Document understanding dataset and evaluation (DUDE).* ICCV (2023).
>
> [5] Li, Kunchang, et al. *MVBench: A comprehensive multi-modal video understanding benchmark.* CVPR (2024).
>
> [6] Liu, Yuanxin, et al. *TempCompass: Do Video LLMs Really Understand Videos?* ACL Findings (2024).

---

> > ### Comment · Reviewer_P1AW · 2025-08-05
> >
> > Appreciate for the detailed response. I think the response have solved my concerns. I changed my rating to accept.

---

> ### Author Response · Authors · 2025-08-05
>
> Thank you for your constructive comments and raised score. We are glad that our responses addressed your concerns, and we will revise the paper following the reviews.
>
> Best regards,
>
> The authors of Paper 15073

---

### Official Review · Reviewer_4xuh · 2025-07-03

**Clarity:** 4
**Significance:** 3
**Originality:** 3
**Rating:** 5
**Confidence:** 3

**Summary:**

The paper proposes PeRL, an RL framework to improve multi-image interleaved vision-language reasoning of VLMs by addressing positional bias and sample difficulty. It is done by (1) introducing image permutation + text rephrasing to learn order-invariant representation and (2) by filtering out easy samples to maintain a balanced training set termed rollout filtering. The training framework is based on GRPO. PeRL is SOTA on multi-image tasks and competitive on single image tasks.

**Questions:**

1. The paper identifies positional bias as a key failure mode of existing VLMs and proposes order-invariant training via permutations as a solution. Could the authors elaborate on why positional invariance is the right solution to this issue, as opposed to, say, explicitly modeling position-aware visual references or learning to attend over inter-image relationships in a more structured way?

2. As a follow-up, while the proposed permutation-based augmentation improves order invariance, have the authors explored robustness to more complex variations such as occlusions, missing images, or distractors? Would the method still be effective in such cases, or is it narrowly targeted to positional shuffling?

**Ethical Concerns:**

["NO or VERY MINOR ethics concerns only"]

**Final Justification:**

The authors' response answered all of my concerns. I am happy with their response. I accept the paper.

**Quality:**

4

**Strengths And Weaknesses:**

Strengths:
- Well motivated. The paper clearly introduces an issue with existing VLMs and makes intuitive design choices in the training to solve them.
- Good empirical improvement across multi-image benchmarks without sacrificing other aspects (single image reasoning)

Weaknesses:
- The core idea of simply permuting image order and rephrasing the text is relatively simple. I do not think it can scale well to complex real-world situations requiring hierarchal reasoning.

---

> ### Author Rebuttal · Authors · 2025-07-31
>
> Dear Reviewer 4xuh,
>
> We sincerely thank you for your time and your constructive and insightful review. We are very encouraged that you praise our work as being **well-motivated** with intuitive design choices (S1) and for achieving SOTA empirical improvements on multi-image benchmarks **without sacrificing single-image reasoning performance (S2)**. Below, we provide detailed responses to your concerns.
>
>
> ---
>
> ### **W1: The core idea of simply permuting image order and rephrasing the text is relatively simple. I do not think it can scale well to complex real-world situations requiring hierarchical reasoning.**
>
> **1. Simple yet effective.**   We agree that the core concept is simple and intuitive, but we view this as a key strength of our approach. PeRL builds on the theoretically and empirically validated GRPO framework, inheriting its reliability without altering core reasoning mechanics. This design enables our model to maintain competitive performance on demanding single-image math benchmarks, even against specialized models like **MMEureka** [1] and **R1-OneVision** [2].
>
> Our work demonstrates that it is possible to improve positional invariance while simultaneously **preserving and strengthening reasoning abilities**. This balance confirms that our “simple” yet principled approach is effective and scalable for increasingly complex tasks.
>
> **2. Performance in more complex real-world situations.**  Addressing positional bias is, in our view, a **prerequisite for reliable higher-level reasoning**. A model cannot exhibit robust hierarchical reasoning if its predictions are sensitive to arbitrary input permutations. By first teaching the model to be **order-invariant**, we ensure it consistently integrates evidence from all images, establishing the conceptual foundation for more advanced compositional and hierarchical reasoning.
>
> Our SOTA results across diverse multi-image and single-image benchmarks empirically support the effectiveness of this straightforward yet foundational correction.
>
> **Additional experiments on complex scenarios.** Inspired by Reviewer P1AW’s feedback, we conducted further experimentsto evaluate PeRL’s generalization to more **realistic, challenging settings**:
> - **SlideVQA** [3]: up to 20 images per item (slide understanding)
> - **DUDE** [4]: up to 120+ images (document understanding)
> - **MVBench** [5] and **TempCompass** [6]: video understanding tasks
>
> | Model          | SlideVQA ANLS | SlideVQA Exact Match | SlideVQA F1 Score | DUDE ANLS | MVBench ACC | TempCompass ACC |
> |----------------|---------------|----------------------|-------------------|-----------|-------------|-----------------|
> | Qwen-VL-2.5 7B | 56.9          | 43.9                 | 51.1              | 60.2      | 61.9        | 68.8            |
> | +PeRL          | **59.2 (+2.3)**   | **46.0 (+2.1)**          | **53.2 (+2.1)**       | **65.5 (+5.3)** | **64.5 (+2.6)** | **70.0 (+1.2)**     |
>
> We consider video understanding to be an even more challenging and hierarchical reasoning form of multi-image reasoning, since it requires analyzing semantic and relational cues across frames.
>
> These findings demonstrate that PeRL not only excels on prior benchmarks but generalizes to more complex real-world settings.
>
>
> ---
>
> ### **Q1: Why is positional invariance the right solution to this issue, as opposed to, say, explicitly modeling position-aware visual references or learning to attend over inter-image relationships in a more structured way?**
>
> This is an excellent question that allows us to clarify our motivation. Our method targets a specific, prevalent failure mode in VLMs: exploiting positional shortcuts when image order is semantically irrelevant to the task.
>
> We considered alternatives such as explicitly modeling position-aware references. For example, post-hoc attention correction methods like **SoFA** [7] can mitigate bias. However, their benefits **diminish as models become stronger**, and they function more as temporary fixes rather than addressing the **underlying positional bias introduced during pre- or post-training**.  This may stem from the fact that their architectures are specifically designed for the multi-image setting; lacking targeted training, they exhibit weaker generalization to more complex reasoning tasks.
>
> We found that modifying attention mechanisms in this way sometimes **degrades single-image reasoning performance**, suggesting that our principled permutation approach is a more stable and generalizable solution.
>
> | Model                 | BLINK       | Mantis-Eval  | MathVista    | MathVision   |
> |-----------------------|------------|-------------|-------------|-------------|
> | LLaVA-NeXT+SoFA       | 54.9 (+2.5) | 61.23 (+2.1) | 34.2 (+0.2) | 13.8 (+0.0) |
> | Qwen-2.5-vl-7B+SoFA   | 57.5 (+2.3) | 71.4 (+0.6)  | 66.3 (-1.9) | 25.0 (-0.1) |
> | Qwen-2.5-vl-7B+PeRL   | **58.5 (+3.3)**| **76.4 (+5.6)** | **73.0 (+4.8)** | **28.3 (+3.1)** |
>
>
>
> ---
>
> ### **Q2: Have you explored robustness to more complex variations such as occlusions, missing images, or distractors? Would the method still be effective in such cases, or is it narrowly targeted to positional shuffling?**
> Thank you for this in-depth question. We agree that robustness to such variations is a crucial long-term goal.
>
> We argue that **mitigating the bias is a foundational step**, enabling the model to better learn inter-image relationships, not just memorize positional patterns, which is essential for handling more complex scenarios.
>
> Although most benchmarks do not include these variations, we performed a **preliminary experiment**:
> - Sampled 1k random items from the BLINK dataset
> - Simulated a “missing image” scenario by replacing images in some answer options with “None of the answers are correct”
>
> On this modified set, accuracy improved from **50.4% to 52.3%**.
> We attribute this improvement to two factors: first, for some questions, removing an incorrect option reduces difficulty by eliminating a distractor. Second, and more importantly, it indicates that our model's core understanding of inter-image relationships has genuinely improved, allowing it to better handle unexpected changes in the input. This suggests our method provides a solid foundation for robustness against more complex variations.
>
> This indicates that our method lays a strong foundation for further robustness against even more complex variations in future research.
>
>
> ---
>
> ### **References**
>
> [1] Meng, Fanqing, et al. Mm-eureka: Exploring the frontiers of multimodal reasoning with rule-based reinforcement learning. arXiv preprint arXiv:2503.07365 (2025).
>
> [2] Yang, Yi, et al. R1-onevision: Advancing generalized multimodal reasoning through cross-modal formalization. arXiv preprint arXiv:2503.10615 (2025).
>
> [3] Tanaka, Ryota, et al. SlideVQA: A dataset for document visual question answering on multiple images. Proceedings of AAAI 37.11 (2023).
>
> [4] Van Landeghem, Jordy, et al. Document understanding dataset and evaluation (DUDE). ICCV (2023).
>
> [5] Li, Kunchang, et al. MVBench: A comprehensive multi-modal video understanding benchmark. CVPR (2024).
>
> [6] Liu, Yuanxin, et al. TempCompass: Do Video LLMs Really Understand Videos? ACL Findings (2024).
>
> [7] Tian, Xinyu, et al. Identifying and mitigating position bias of multi-image vision-language models. CVPR (2025).

---

> > ### Comment · Reviewer_4xuh · 2025-08-04
> >
> > Thank you for your detailed response. The clarifications on order invariance and the added experiments are helpful and appreciated.
> >
> > A minor suggestion: while the missing-image experiment is small but a nice touch, a more systematic robustness analysis (e.g., with distractors or occlusions) would strengthen the generalization claim. More directly acknowledging limitations of your current scope in the paper (e.g., "our method is _not yet_ **explicitly** designed to handle occlusion or distractors") in the discussion section would be amazing!
> >
> > My confidence in the paper has improved post-rebuttal. I am leaning towards accept.

---

> > > ### Author Response · Authors · 2025-08-05
> > >
> > > Thank you for your constructive feedback and positive evaluation. We are glad that our Rebuttal has received your approval, and your suggestions are also very important. We will incorporate your opinions and make modifications in further versions.
> > >
> > > Best regards,
> > >
> > > The authors of Paper 15073

---

### Decision · Program_Chairs · 2025-09-17

**Decision:**

Accept (poster)

**Comment:**

The paper proposes a PeRL method for interleaved multimodal tasks, which uses image sequence permutation to increase spatial and positional diversity, and a rollout filtering mechanism for resampling to focus on important trajectories.

The original main concerns from the reviewers include:

- may not scale well to more complex tasks and more images (4xuh, P1AW)
- unclear details of the method such as the rule-based reward, data processing (P1AW, HKDU)
- lack of evaluation of data preprocessing from GPT4o (P1AW, rHiB)
- computational cost (HKDU)
- theoretical analysis (rHiB)

The rebuttal addressed most of these concerns, and thus reviewers have increased scores to 2 Accepts and 2 Borderline Accepts. The authors should update the paper according to the reviewers' comments in the final version.